# Capturing the Denoising Effect of PCA via Compression Ratio

**Chandra Sekhar Mukherjee** *
chandrasekhar.mukherjee@usc.edu

**Nikhil Deorkar** *
deorkar@usc.edu

**Jiapeng Zhang** *
jiapengz@usc.edu

## Abstract

Principal component analysis (PCA) is one of the most fundamental tools in machine learning with broad use as a dimensionality reduction and denoising tool. In the later setting, while PCA is known to be effective at subspace recovery and is proven to aid clustering algorithms in some specific settings, its improvement of noisy data is still not well quantified in general.

In this paper, we propose a novel metric called *compression ratio* to capture the effect of PCA on high-dimensional noisy data. We show that, for data with *underlying community structure*, PCA significantly reduces the distance of data points belonging to the same community while reducing inter-community distance relatively mildly. We explain this phenomenon through both theoretical proofs and experiments on real-world data.

Building on this new metric, we design a straightforward algorithm that could be used to detect outliers. Roughly speaking, we argue that points that have a *lower variance of compression ratio* do not share a *common signal* with others (hence could be considered outliers).

We provide theoretical justification for this simple outlier detection algorithm and use simulations to demonstrate that our method is competitive with popular outlier detection tools. Finally, we run experiments on real-world high-dimension noisy data (single-cell RNA-seq) to show that removing points from these datasets via our outlier detection method improves the accuracy of clustering algorithms. Our method is very competitive with popular outlier detection tools in this task.

## 1 Introduction

Principal component analysis, commonly known as PCA, is one of the most fundamental tools in machine learning. PCA is primarily used as a dimensionality reduction tool that transforms high-dimensional data to lower dimensions for better visualization as well as a heuristic that reduces the complexity of the algorithms that are to be run on the data. On the other hand, it is also known to have certain denoising effects on high-dimensional data. This denoising phenomenon has been observed in different domains, including biological data [KAH19, VKS17], speech data [Li18], signal measurement data [ARS+04, KHK19], image data [MBSP12] among others. The denoising effect of PCA has been extensively studied over the last decades [And58, HR03, Jac05, RVdBB96, Nad08, Nad14, VN17, MZ23, MZ24].

---

*Thomas Lord Department of Computer Science, University of Southern California. Research supported by NSF CAREER award 2141536.

38th Conference on Neural Information Processing Systems (NeurIPS 2024).

One of the most fundamental problems in unsupervised learning is the analysis of data in the presence of community structures [CA16]. This includes clustering of such data [XT15], visualization [TWT21], outlier detection [AM13], and others. The primary progress in understanding the denoising effect of PCA has been solely in clustering, particularly in connection to the K-Means algorithm [DH04, KK10, AS12], where PCA in combination with a K-Means based iterative algorithm is shown to provide a good clustering of that dataset with mild assumptions with the underlying community structure.

However, PCA seems to have a more "general" denoising effect in data, as it improves the performance of various downstream algorithms, including clustering [VKS17] as well community structure preserving graph embedding [HHAN$^+$21] and this denoising effect is evident in many real-world datasets.

## 1.1 Contributions

To this end, we propose a metric, called *compression ratio*, that quantifies PCA's improvement of high dimensional noisy data with underlying community structure in a geometric, and thus algorithm-independent manner [2].

**Compression ratio.** Let $u$ and $v$ be two data points from a dataset and let $\Pi_t$ be the $t$-dimensional PCA projection operator. Then the compression ratio between the two points is defined as the ratio between their pre-PCA and post-PCA distance, which is

$$\frac{\|u - v\|}{\|\Pi_t(u) - \Pi_t(v)\|}.$$

In a dataset with a community structure, we show the compression ratio for intra-community pairs is higher than that of inter-community pairs even in settings where the pre-PCA inter-community and intra-community distances are very similar. We demonstrate (through a *random vector mixture model*) that this ratio gap reflects the denoising effect of PCA. As a consequence, PCA brings points from the same community much closer, improving the performance of downstream algorithms such as K-Means.

As a motivating byproduct, we show that this metric can be used to design an outlier detection method that can detect points deviating from a community structure. Furthermore, we show that this method can improve the accuracy of clustering algorithms in real-world high-dimensional datasets.

**Outlier detection method.** Our outlier detection is a simple process inspired by compression ratio. Intuitively, any data point that belongs to an underlying community should have large compression ratios with many points from the same community, whereas it will have a lower compression ratio w.r.t inter-community points. On the other hand, outliers will have more similar compression ratios with all the other points. This difference can be captured by the variance of the list of compression ratios between one point and all of the other points, with outliers having a lower variance of compression. Thus our algorithm simply removes points with low variance of compression.

We analyze this simple algorithm in an extension of the standard random vector mixture model. We also compare our algorithm with popular algorithms such as the Local Outlier Factor (LOF) method [BKNS00] and KNN-dist [RRS00] as well as more recent methods such as Isolation forest [LTZ08] and ECOD [LZH$^+$22] through both simulations and experiments on real-world data. We show that this simple algorithm is very competitive with those popular outlier detection tools.

Overall, we believe the effect of PCA on denoising becomes more significant if for each datapoint, there are many data points with large compression ratio variance.

**Real world experiments** Finally, we test the relevance of compression ratio as a metric and the outlier detection method in real-world data. We focus on single-cell data, as it is both high dimensional $(20,000 - 40,000$ dimension) and noisy [KAH19], using datasets from a popular benchmark database [DRS20] with ground truth community labels. We first show that the average intra-community compression ratio is higher than the average inter-community compression ratio

---

[2]From hereon, we use the word community to refer to the underlying structure of the data, whereas clustering of data refers to the outcome of a particular clustering algorithm on the dataset.

in all of the datasets. We then show that removing outliers in these datasets via our variance of compression technique improves the performance of clustering algorithms, such as PCA+K-Means, where we again outperform standard outlier detection methods.

**Organization of the paper**  In Section 2, we discuss our theoretical analysis. Concretely, we define the random vector mixture model and provide bounds on the intra-community and inter-community compression ratios. Next, we define our outlier detection metric and justify it in an extension of our generative model. Section 3 contains the simulation results validating the compression ratio metric and we also compare the performance of our outlier detection method with other methods. Finally in Section 4 we demonstrate that PCA exhibits an average version of compression ratio in real-world biological datasets [DRS20] and then test our outlier detection-based clustering accuracy improvement idea discussed above.

## 1.2   Related Works

PCA and its effect on noisy data has been subject to a lot of investigation in the last 50 years. Before 2008, most of the work focused on the asymptotic setting, where the number of points ($n$) and/or the dimension ($d$) are infinite (see [And58, RVdBB96, HR03, Jac05] and the references therein). In the last two decades, several works have also considered the finite sample setting [Nad08, Nad14]. These works have primarily focused on the denoising aspect of PCA in different variants of Gaussian noise. In a recent line of work [VN17] has studied the subspace recovery problem in the presence of bounded and (nice) sub-Gaussian noise. However, there seems to be no direct way to convert these results into a clustering setting. In comparison, we study PCA's denoising effect on data in random vector mixture model via the compression ratio metric, where the noise can be heavy sub-Gaussian.

**PCA in clustering tasks**  With regards to PCA's impact on data with community structure, the primary work has been in connection to K-Means. Here, one of the first works [DH04] showed that the outcome of PCA can be viewed as an approximation result to the K-Means outcome in clustering data. In this direction, a lot of progress has been made in the last two decades.

A beautiful recent work [KK10] has shown that PCA followed by several iterations of K-Means along with modifications can cluster data with reasonable parameters in the random vector mixture model that we discuss here, which was then improved in [AS12]. Both of the works focused on the setting of $n \gg d$ (for example, [KK10] worked with $n \geq d^8$). More recently, tighter results have been obtained in the context of the Gaussian-mixture model in [LZZ21] (still on the setting of $n \gg d^2$).

In comparison, we study PCA's relative compression in an algorithm-independent fashion, focusing on its effect on the geometry of the data in the high-dimensional setting of $n = \Omega(d)$ with sub-Gaussian noise. We are motivated to analyze this setting as single-cell datasets often have $n < d$.

## 2   Random vector model and relative compression of PCA

To theoretically study the relative compression of PCA, we use a high-dimensional mixture model, similar to one in [KK10, AS12]. We call this a random vector mixture model. This can also be interpreted as a signal-plus-noise model where the signal imposes a community structure on the data. The dataset of interest is a set of $n$ many $d$ dimensional real vectors $\mathbf{x_i} \in \mathbb{R}^d, 1 \leq i \leq n$, which is together represented as the dataset $X$. We express $X$ as a $d \times n$ matrix, with each column representing a data point. The data points have an underlying hidden community structure that is expressed as a partition of $[n] := \{1, \ldots, n\}$ into $k$ many sets $V_1, \ldots, V_k$ such that each $i \in [n]$ lies in any one $V_j$. We then have the following problem structure.

1. Each cluster $V_j, 1 \leq j \leq k$ is associated with a ground truth center $\mathbf{c_j} \in \mathbb{R}^d$.

2. Additionally, each cluster $V_j$ is associated with a distribution $\mathcal{D}^{(j)}$ such that $\mathcal{D}^{(j)}$ is a *coordinate wise independent zero mean* distribution. For ease of exposition, we define the support of $\mathcal{D}^{(j)}$ to be $[-\alpha, \alpha]^d$ for some $\alpha$ (which can depend on $n, d$), but our methods also directly apply to sub-Gaussian distributions where each coordinate has a constant sub-Gaussian norm (resulting in $\mathcal{O}(\sqrt{d})$ norm of any column vector). Then $\alpha$ would be replaced with $C' \log n$ for some constant $C'$ in our bounds.

Then, the dataset $X$ is set up as follows.

**Definition 2.1** (Random vector mixture model)**.** For each $i \in [n]$, if $i \in V_j$, then $\boldsymbol{x_i} = \mathbf{c_j} + \boldsymbol{e_i}$ where $\boldsymbol{e_i} \sim \mathcal{D}^{(j)}$, i.e. $\boldsymbol{e_i}$ is independently sampled from $\mathcal{D}^{(j)}$. Here we abuse notation and denote both $i \in V_j$ as well as $\boldsymbol{x_i} \in V_j$.

With this setup, now we define the PCA projection operator and the compression ratio metric formally.

**Definition 2.2** (The PCA operator $\Pi_X^{k'}$)**.** Let $X$ be a $d \times n$ matrix. Then the $k'$ dimensional PCA projection operator is simply the projection operator onto the first $k'$ principal components of $X$.

Next we formally define the compression ratio metric.

**Definition 2.3.** For any pair $(i, i')$ we define the $k'$-PC compression of the pair of vectors in $X$ as

$$\Delta_{X,k'}(i, i') = \frac{\|\boldsymbol{x_i} - \boldsymbol{x_{i'}}\|}{\|\Pi_X^{k'}(\boldsymbol{x_i}) - \Pi_X^{k'}(\boldsymbol{x_{i'}})\|}$$

Before describing our results, we define certain parameters of the model.

1. The maximum variance of the entries, $\sigma$ is defined as $\sigma^2 = \max_{1 \leq j \leq j, 1 \leq \ell \leq d} \mathrm{Var}[\mathcal{D}_\ell^{(j)}]$

2. The average variance of a column in a distribution $\mathcal{D}^{(j)}$, noted as $\sigma_j$ is defined as $\sigma_j^2 = \frac{1}{d} \sum_\ell Var([\mathcal{D}_\ell^{(j)}])$. Here, $\sigma_j \sqrt{d}$ is the perturbation on the data points of $V_j$ due to the noise.

In this direction, we first lower, and upper bound the $(k-1)$-PC intra-community and inter-community compression ratios respectively, as a function of the maximum variance, average variances, spectral structure of the noise and signal, and distance between the centers of the model, which can be found in Theorem B.1.

Although our result applies to any set of centers, the spectral properties of the resultant matrix, and their interactions make the result hard to interpret. To give more insight into our bounds, we instead define a restricted (still natural) structure on the centers, which allows us to give a more interpretable result in this case. For simplicity, we also work in the setting where $d \geq 10\alpha\sqrt{n} \log n$.

**Definition 2.4** (Spatially unique centers)**.** We say a set of vectors $\mathbf{C} = \{\mathbf{c_1}, \ldots, \mathbf{c_k}\}$ are $\gamma$-spatially unique, if we have that

$$\min_{1 \leq j \leq k} \min_{\mathbf{v} \in \mathsf{Span}(\mathbf{C} \backslash \mathbf{c_j})} \|\mathbf{c_j} - \mathbf{v}\| \geq \gamma$$

This implies that each center has a unique pattern that cannot be approximated by a combination of the other centers. Here note that $\gamma \geq \min_{j \neq j'} \|\mathbf{c_j} - \mathbf{c_{j'}}\|$. For example, such a property is expected if the centers are mutually orthogonal. One can also think of them as vertices in a high-dimensional regular polygon. Then, we give some sufficient conditions for the separation of intra-community and inter-community compression ratios of PCA.

**Theorem 2.5** (Separation of compression ratio)**.** *Let $\gamma \geq C \max\{\sigma\sqrt{k}d^{1/4}, \sigma\sqrt{k} + \alpha \log n\}$ for some constant $C$. Furthermore, let $i \sim i'$ denote that $\boldsymbol{y_i}$ and $\boldsymbol{y_{i'}}$ belong to the same underlying community. Then, the following holds.*

1. *The perturbation of the points due to noise can be much larger than the distance between the community centers, i.e., the noise dominates the distance between the centers.*

2. *With probability $1 - \mathcal{O}(1/n)$, $\min_{(i,i'):i \sim i'} \Delta_{X,k-1}(i, i') \geq 4 \cdot \max_{(i,i'):i \nsim i'} \Delta_{X,k-1}(i, i')$*

This shows that the compression ratio of PCA provides a separation between intra-community and inter-community pairs even in a setting where the noise highly dominates the distance between the centers. One can find a more general theorem w.r.t spatially unique centers in Theorem C.4.

A natural question is whether post-PCA distance is a good metric for denoising due to PCA. In this regard, we point out that the compression ratio has an added normalization property. For example, consider the case where all pair-wise center distances are the same. In such a case, the post-PCA distances are dependent on $\sigma_j$, so communities with larger variances have larger intra-community distances. However, this gets normalized in the compression factor as per Equation (9) of Theorem C.4, as the numerator also has a dependency on $\sigma_j$.

---

**Algorithm 1** Outlier detection via variance of compression ratio

---

**Input:** data $X$, PCA dimension $k'$, number of outliers $o$.
**for** $i = 1$ **to** $n$ **do**
   $SC[i] \leftarrow \mathsf{VAR}\Delta_{X,k'}(\boldsymbol{x_i})$ {$\mathsf{VAR}\Delta_{X,k'}$ defined in Eq. 1}
**end for**
**return** $o$ many indexes with lowest $SC$ values.

---

## 2.1 Outlier detection with compression ratio

Now, we discuss the usefulness of compression ratio on outlier detection. We first describe the notion of variance of compression ratio.

**Definition 2.6** (Variance of compression ratio). Given a dataset $X$ and a PCA dimension $k'$, variance of compression ratio of a point $\boldsymbol{u} \in X$ is defined as

$$\mathsf{VAR}\Delta_{X,k'}(\boldsymbol{x_i}) = \mathrm{Var}(\{\Delta_{X,k'}(i, i')\}_{i' \neq i}) \tag{1}$$

where $\Delta_{X,k'}(i, i') = \frac{\|\boldsymbol{x_i} - \boldsymbol{x_{i'}}\|}{\|\Pi_X^{k'}(\boldsymbol{x_i}) - \Pi_X^{k'}(\boldsymbol{x_{i'}})\|}$ is the compression ratio between points $i$ and $i'$.

That is, it is simply the variance of the list of compression ratios of $\boldsymbol{x_i}$ with all the other points $\boldsymbol{x_{i'}}$.

Then, our intuition is that if data consists of many points from the high dimensional mixture model, as well as several outlier points that don't share a common signal (center), they have a lower variance of compression ratio. We concretize this notion with the following simple detection algorithm 1.

**Mixture-model with outliers**  Now let us consider an extension of the mixture model in Definition 2.1 to incorporate outliers.

**Definition 2.7** (Mixture model with outliers). Let $X$ be a $d \times n$ dataset with the partition $V_1, \ldots, V_k, \hat{V}$, a set of $k$ centers $\{\mathbf{c_j}\}_{j=1}^k$ and distributions $\{\mathcal{D}^{(j)}\}_{j=1}^k + 1$ with the following generation method.

1. *clean points:* If $i \in V_j, 1 \leq j \leq k$, $\boldsymbol{x_i} = \mathbf{c_j} + \boldsymbol{e_i}$ where $\boldsymbol{e_i}$ is sampled from $\mathcal{D}^{(j)}$.

2. *outliers:* If $i \in \hat{V}$, then we sample $p_{i,1}, \ldots p_{i,k} \in [0.5, 1]$. Then $\boldsymbol{u_i} = \sum_j \alpha_{i,j} \mathbf{c_j} + \boldsymbol{e_i}$ where $\alpha_{i,j} = \frac{p_{i,j}}{\sum_j p_{i,j}}$ and $\boldsymbol{e_i}$ is sampled from $\mathcal{D}^{(k+1)}$.

Let $|\hat{V}| = n_o$ and $n = n_o + n_c$. To keep the results simple, we make the average variance of each distribution $\mathcal{D}^{(j)}$ same, which is $\sigma'$.

Such a scenario can occur in many different settings. For example, consider single-cell datasets which is a popular biological data type. Here each data point is a cell and the features (which are high, such as $20,000$) are specific gene expressions, A primary task here is to obtain cell sub-populations [THL$^+$19, VKS17, KAH19]. Although the gene expressions within sub-populations should have similarities, they are perturbed by biological and technical noise, making high-dimensional mixture models a good setup to study them. However, some cells may not belong to any particular sub-populations, but rather be intermediate cells. Additionally, sometimes cells get merged during the biological experiment that records the gene expressions, generating data points that behave like a random mixture of two or more data points. Our model aims to model such scenarios.

In this setting, we get the following outlier detection result where the centers have spatially unique centers.

**Theorem 2.8** (Outlier detection via Algorithm 1). *Let $X$ be a $d \times n$ dataset with $\gamma$-spatially unique $k$ many centers where $\log n \leq k \leq \sqrt{d}$ and $n_0$ outliers in the setting of Definition 2.7. Let the following conditions hold*

1. $\|\mathbf{c_j} - \mathbf{c_{j'}}\| = \mathcal{O}(\sigma'\sqrt{d})$   *(the noise is significant)*
2. $\gamma \geq 2C'\sigma^{3/2}/\sigma' \cdot k \cdot d^{1/4} \log n$

*Then, for any $n_0 \le n/2$, the first $n_0$ points ranked by Algorithm 1 all belong to the outlier group ($\hat{V}$) with probability $1 - o(1)$.*

We discuss the connection between our results and the role of spatially unique centers in Appendix C. The proof of Theorems 2.5 and 2.8 can be found in Appendix C and C.2 respectively.

This gives us initial theoretical evidence that in the random-mixture model with outliers, our simple outlier detection method can detect outliers when a non-negligible fraction of the points are outliers. Next, we use simulations of our model to test the efficacy of our outlier detection method and its impact on the community structure of data and compare them with some popular outlier detection methods.

## 3 Simulations for outlier detection

In this section, we first describe different instantiations of the random-vector mixture model, observe the intra-community and inter-community compression ratios in them, and then run simulations in the outlier mode. All simulations and experiments were run on a 2020 M1 MacBook Pro with 16 GB of memory within 1.5 hours of total running time.

**Simulation setup**    For this setup, we set $n = 3000, d = 1000$, and $k = 3$, with each community having the same number of points. For simplicity, we choose 3 equidistant centers, with $\|\mathbf{c_i} - \mathbf{c_j}\| = \mathbf{c}$. We set the noise distributions to be Bernoulli distributions with variance $\sigma_1, \sigma_2, \sigma_3$ respectively. We work in two primary settings, of equal and unequal noise.

i) *Equal noise.* Here we have $\sigma_j = \sigma$ for all $i$. ii)*Unequal noise.* Here one of the communities has variance $2\sigma$, whereas all the other communities have variance $\sigma$.

Then, we test the algorithms for three values of $\sigma$ in the following manner.

- *Low noise:* We choose $\sigma : \|\mathbf{c_j} - \mathbf{c_{j'}}\| \approx 3\sigma\sqrt{d}$. This implies distance between the centers dominates the perturbation due to noise.
- *Significant noise:* Here $\sigma : \|\mathbf{c_j} - \mathbf{c_{j'}}\| \approx \sigma\sqrt{d}$. Here the noise norm and distance between centers are of the same order.
- *High noise:* We have $\sigma : \|\mathbf{c_j} - \mathbf{c_{j'}}\| \approx 0.3\sigma\sqrt{d}$. Here the noise heavily dominates the center distances.

Let us look at the equal noise setting, i.e. the case where the variance of noise distributions for all communities are the same. We observe that in the low-noise setting, all intra-community compression ratios are higher than all inter-community compression ratios. As the noise increases, the gap between them decreases, so that in the high-noise setting, there is now an overlap between intra-community and inter-community compression ratios. We demonstrate this in Figure 1a. This further indicates that compression ratio is indeed a useful metric even when the noise has a strong perturbation effect on the data (even though there will be no clean separation between intra-community and inter-community compression ratios once the noise is very high).

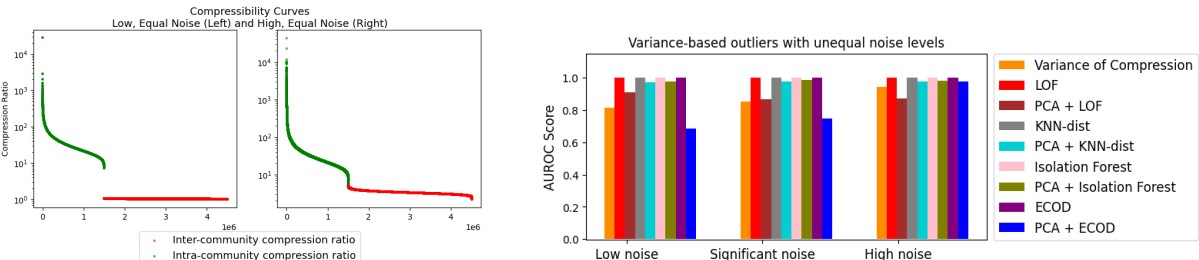

(a) Comparing intra and inter community compression ratios in simulation

(b) AUROC of variance-based outlier removal

Figure 1: Simulation results for compression ratio and outlier detection

### 3.1 Outlier detection

Now, we discuss the outlier detection, starting with the simulation setup in this case. We follow the random-mixture-outlier model and add outlier points along with the clean points as follows.

We add $n_o = n_c/10$ outliers following definition 2.7. That is, we randomly choose $\alpha_1, \ldots, \alpha_3$ and then a random mixture-center is chosen as $\sum_j \alpha_j \mathbf{c_j}$, and then we add a random noise vector from the Bernoulli distribution of variance $\sigma_4$.

**Outlier detection algorithms for comparisons**  Outlier detection has been an active area of study in unsupervised learning, providing several influential algorithms. In a recent, comprehensive benchmarking of outlier detection algorithms, [HHH+22] compared the performance of several unsupervised learning algorithms on different datasets. They found that for unsupervised outlier detection methods, success was related to whether the underlying model of the data assumed by the outlier detection method followed the dataset at hand. They found that for local outliers, the popular Local Outlier Factor (LOF) method [BKNS00] performed the best statistically, whereas for global outliers, KNN-dist (where the outlier score is simply the distance to the $K$-th nearest neighbors) [RRS00] performed the best. Owing to their generally impressive performance, we use them for comparison with our variance of compression method. Furthermore, we select a popular method called Isolation forest [LTZ08] and also a very recent and popular outlier detection method ECOD [LZH+22]. We also use PCA+method for each of the methods as benchmarks, as both outliers and clean points are perturbed by zero-mean noise, and we now understand PCA can help mitigate the effect of said noise, as discussed in Section 2.

**Outlier detection results**  We compare the AUROC values of the 5 outlier methods of interest in these settings. We record the results in Figure 2a and 2b for the equal and unequal noise settings respectively.

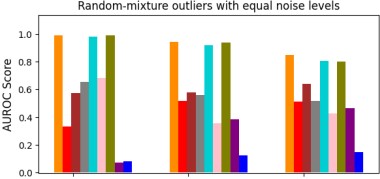 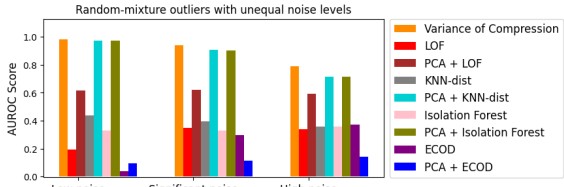

   (a) AUROC of outlier removal in equal noise setting   (b) AUROC of outlier removal in unequal noise setting

Figure 2: Comparison of outlier removal in different noise settings

As we can see, our method results in the highest AUROC value, followed by PCA+KNN-dist. we make two observations.

i) The performance gap between variance-of-compression and the next best method is higher in the unequal noise setting.

ii)As the noise level increases, the gap between our method and PCA+K-NN dist increases.

These two points further highlight the compression ratio's normalizing effect as well as effectiveness in high noise settings.

Here we remark that in real-world data, while some points may indeed behave like outliers, they need not all be the same kind of outlier. Thus, we would like to verify our method's performance in the presence of a different kind of outlier, which we concretize below.

**Higher variance-based outliers**  We consider the case that some points may have significantly higher noise perturbations than others. In this setting, we randomly select some points, and we generate some points with $c \cdot \sigma$ coordinate-wise variance, where $c = 8$ for our experiments (recall that the noise in the other points has a coordinate-wise $\sigma$ variance). It is well known that if noise is low-dimensional, then such outliers are well captured by LOF. We observe that while in the low noise setting our performance is worse than the other methods, as the overall noise increases, the performance of our method is more comparable to the other methods. We record the results in Figure 1b.

| Dataset | Avg. intercluster compression | Avg. intracluster compression |
|---|---|---|
| Koh | 2.539 | 7.468 |
| Kumar | 1.969 | 14.811 |
| Simkumar4easy | 3.577 | 15.808 |
| Simkumar4hard | 5.267 | 15.051 |
| Simkumar8hard | 4.349 | 9.370 |
| Trapnell | 6.373 | 9.857 |
| Zheng4eq | 2.399 | 6.639 |
| Zheng4uneq | 2.215 | 6.260 |
| Zheng8eq | 2.398 | 4.722 |

Table 1: Relative compression on RNA-seq datasets

| Dataset | NMI of PCA + k-means |
|---|---|
| Koh | 0.847 |
| Kumar | 0.924 |
| Simkumar4easy | 0.746 |
| Simkumar4hard | 0.237 |
| Simkumar8hard | 0.449 |
| Trapnell | 0.286 |
| Zheng4eq | 0.690 |
| Zheng4uneq | 0.691 |
| Zheng8eq | 0.554 |

Table 2: Average PCA+K-Means outcome before data removal

This shows that our outlier detection method is adept at detecting different kinds of outliers, outperforming popular outlier detection tools in some settings, and being competitive to them in others. We also observe that as the overall noise in the dataset increases, the performance of our method compared to the other outlier detection tools improves. This further highlights the power of compression ratio when especially dealing with noisy data. Having demonstrated the validity of our outlier detection method in two different settings, across different noise levels, we now focus on real-world datasets.

# 4 Real world experiments

## 4.1 Datasets of interest

In this section, we provide experimental results to exhibit the validity of compression ratio as a metric and the usefulness of our outlier detection method in improving the community structure of datasets. We focus on single-cell RNA sequencing datasets. The dataset consists of $n$ many data points, each corresponding to a cell. The features are gene expressions, and for the cell, the expression of some $d \geq 10,000$ genes are recorded. A fundamental problem here is to identify sub-populations of interest. However, the problem is challenging as the biological process of recording gene expressions is error-prone [THL$^+$19], and gene expressions within the same population may also vary due to internal randomness. Furthermore, experiments can cause cells to get merged during gene-expression recording [XL21]. This makes single-cell RNA sequencing data a good testing ground for high dimensional noisy data with outliers and underlying community structure.

In this direction, we consider the single-cell RNA sequencing datasets from a benchmark paper [DRS20]. These datasets also have moderate to highly reliable ground truth labels, that help us understand the usefulness of our metrics and our algorithm. These datasets vary in the number of cells, genes, clusters, cells per cluster, and the "difficulty" of clustering. A summary of the datasets is provided in Table 4 in Appendix E.1.

## 4.2 Average compression in the datasets

As discussed in Section 2 and described in Theorem 2.5, our primary result showed that the intra-community compression ratios are higher than inter-community compression ratios in a large range of parameters. Here we look at average statistics of compression ratio to provide a first layer of evidence supporting this phenomenon in real-world data. We define the following metric. For any community $V_j$, we define the average intra-community compression ratio as $\mathsf{intra}_{X,k'}(V_j) = \underset{i,i' \in V_j}{\mathbb{E}}\left[\frac{\|x_i - x_{i'}\|}{\|\Pi_X^{k'}(x_i - x_{i'})\|}\right]$ Similarly, the average inter-community compression ratio is defined as $\mathsf{inter}_{X,k'}(V_j) = \underset{i \in V_j, i' \in [n] \setminus V_j}{\mathbb{E}}\left[\frac{\|x_i - x_{i'}\|}{\|\Pi_X^{k'}(x_i - x_{i'})\|}\right]$. This gives an average measurement of the compression ratio in the dataset. In this regard, we find that for each of the 9 datasets and each of the communities in the dataset, the intra-community compression ratio is higher than the inter-community compression ratio. We provide the results in the Appendix E.2. Here, for brevity we present the average of $\mathsf{intra}_{X,k-1}(V_j)$ and $\mathsf{inter}_{X,k-1}(V_j)$ for each dataset in Table 1.

### 4.3 Improvement of clustering results via outlier detection

Next, we study the usefulness of our outlier detection method in these datasets. Unlike our simulations, there is no ground-truth labeling for outliers. Rather, we assume that in each community (as defined by the labels provided with the dataset), some points may behave more like an outlier, in that they may be a mixture of different signals. These can also be points that have uncharacteristically high noise compared to the rest of the data points. In such a case, these data points may muddle the community structure in the datasets, and thus, removing them may improve the community structure of the datasets. We capture this improvement by observing the change in the accuracy of clustering algorithms when some outlier-like points are removed from the dataset. For our experiments, we choose PCA+K-Means as our clustering algorithm, as it is known to be effective in single-cell datasets [VKS17, KAH19].

**Experimental setup** For each of the datasets, we do the following. Let $k$ be the number of communities. We apply some $c$-dimensional PCA and then run a standard implementation of K-Means with $k$-centers on the post-PCA data and record the NMI and purity score, which are popular clustering accuracy metrics. This gives us a starting point. Then, for each dataset, we apply 9 outlier detection methods. The algorithms are our variance-of-compression-ratio method, and the original and PCA-added versions of LOF, KNN-dist, Isolation forest, and ECOD. We have two settings.

First, we set $c = k - 1$ (following our theory), and obtain the initial PCA+K-means results in Table 2. Then, we remove $5\%$ of the points according to the outlier score and then run PCA+K-Means on the rest of the dataset and obtain the new NMI values. Next, we repeat the same experiments by removing $10\%$ of the points. Additionally, we also use $c = 2k$, and there, calculate the outcome only for $10\%$ points removal, primarily to reduce redundancy. This is to test the sensitivity of the methods to the choice of PCA dimension.

**Results** As a comprehensive summary, we calculate the performance rank of the methods on all the datasets in each of the settings. We record the results in Table 3. As can be observed, we obtained the best rank in *5 out of 6 settings*. The performance of each method for each dataset in the settings can be found in Appendix E.

| Algorithm | Average Rank | | | | | |
|---|---|---|---|---|---|---|
| | NMI, dim = k - 1, 5% removal | NMI, dim = k - 1, 10% Removal | Purity, dim = k - 1, 5% Removal | Purity, dim = k - 1, 10% Removal | NMI, dim = 2k, 10% Removal | Purity, dim = 2k, 10% Removal |
| Var. of Compression | **2.333** | **2.333** | 3.444 | **2.111** | **2.889** | **2.556** |
| LOF | 4.222 | 5.0 | 5.667 | 5.444 | 5.333 | 6.556 |
| PCA + LOF | 3.556 | 4.0 | 4.556 | 4.222 | 4.222 | 5.667 |
| KNN | 5.0 | 4.333 | 3.111 | 3.556 | 3.778 | 3.444 |
| PCA + KNN | 4.556 | 4.556 | 3.778 | 3.778 | 4.667 | 4.333 |
| Isolation Forest | 4.667 | 6.0 | 4.333 | 5.111 | 5.667 | 5.222 |
| PCA + Isolation Forest | 4.222 | 4.333 | **2.444** | 3.111 | 4.444 | 2.667 |
| ECOD | 4.889 | 3.556 | 3.556 | 3.111 | 3.667 | 3.0 |
| PCA + ECOD | 6.333 | 5.111 | 4.111 | 4.667 | 4.556 | 4.222 |

Table 3: Average rank of improvement across all algorithms and experimental settings

**Robustness to choice of dimension** Finally, we note that the compression ratio is not overly sensitive to the choice of PCA dimension, and if we use more dimensions than the number of communities, we still get favorable results. For theoretical support, we show in Section E.4 of the appendix that the compression ratios of most points change only mildly when $k' > k$. In terms of experiments, we verify it as follows. For $k' = 2k$, we calculate the average intra-community and inter-community compression ratios in Appendix E.4 and find them to be consistent with Table 1. As in the case with PCA dimension$= k - 1$, our methods have the best performance in terms of improving clustering performance.

**Limitations** Finally, we note a few limitations with our outlier removal algorithm. First, the algorithm is dependent on selecting a reasonable removal percentage. While we observed greater NMI improvement with greater removal rates, it is important to understand what is a suitable choice for different datasets. Another concern is that our outlier detection tool may not be optimal for handling highly unbalanced communities, as a very small community will show a lower variance of compression ratio. These remain interesting research directions. We note more future directions in the Appendix F.

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

# A    Organization of the appendix

In Sectiion B we obtain our first, generic proofs for compression ratio. Next, in Section C we interpret our results through the lens of spatially unique centers, and also prove our variance of compression result on outlier detection in this setting. Next in Section D we extend the results of Section B when number of components is more than $k + 1$.

Section E contains continuation of experimental results from the main paper. We conclude with some future directions in Section F.

# B    Primary theorem and proof

In this section, we describe our primary compression ratio related result in the random vector mixture model. We first describe our result when the projection dimension is $k - 1$. We first define some notations and useful results that we will use.

## B.1    Preliminaries

We first define the SVD projection operator for a matrix $X$. Let the $k'$-dimensional SVD projection operator for a matrix $X$ be $P_X^{k'}$.

Next, for the dataset matrix $X$, we denote by $Y$ its centered version. Then we have $\Pi_X^{k'} = P_Y^{k'}$.

Then the compression ratio of the data pair $(i, i')$, defined as $\frac{\|x_i - x_{i'}\|}{\|\Pi^{k'}(x_i - x_{i'})\|}$ is in fact $\frac{\|y_i - y_{i'}\|}{\|P_Y^{k'}(y_i - y_{i'})\|}$. Then we have the following bound on the compression ratios in the random vector mixture model.

**Theorem B.1** (Main result). *Let $X$ be a $d \times n$ dataset setup in the random vector mixture model with $k$ underlying communities, so that all centers $\mathbf{c_j}$ and all column vectors $x_i \in X$ are in $[-\alpha, \alpha]^d$. Let $Y$ be the corresponding centered dataset. Considering the following notations,*

1. *$\delta_{k'}(M) := s_{k'}(M) - s_{k'+1}(M)$ for any $M$,*

2. *$\sigma^2$ be the maximum variance of the random variables,*

3. *$\mathcal{N} := C_0 \sigma \sqrt{d + n}$ for some constant $C_0$*

*If $\sigma^2 \geq C_1 \frac{\log n}{n}$ for some constant $C_1$ then with probability $1 - \mathcal{O}(1/n)$ we have that the $(k-1)$-PC compression ratio, $\Delta_{X,k-1}(i, i')$ of* all intra-cluster *pairs $(i, i')$ is lower bounded as*

$$\Delta_{X,k-1}(i, i') \geq$$
$$\frac{\sqrt{2d\sigma_j^2 - 12\alpha\sqrt{d}\log n}}{2\sqrt{2}\left(\sigma\sqrt{k} + C_1 \cdot \alpha \cdot \sqrt{\log n} + \frac{2\mathcal{N} \cdot \sqrt{\sigma_j^2 d + 12\sqrt{d}\log n}}{\delta_{k-1}(Y)}\right)} \tag{2}$$

*Similarly, the compression ratio of all inter-cluster pairs (i,i') is upper bounded by*

$$\Delta_{X,k-1}(i, i') \leq$$
$$\frac{\sqrt{d(\sigma_j^2 + \sigma_{j'}^2) + \|c_j - c_{j'}\|^2 + 12\alpha\sqrt{d}\log n}}{\sqrt{2}\left(\|c_j - c_{j'}\| - 2\left(\sigma\sqrt{k} + C_1 \cdot \alpha \cdot \sqrt{\log n} + \frac{2\mathcal{N} \cdot \sqrt{\|c_j - c_{j'}\|^2 + 2d\sigma^2 + 12\sqrt{d}\log n}}{\delta_{k-1}(Y)}\right)\right)} \tag{3}$$

*with probability $1 - \mathcal{O}(1/n)$.*

Here we make the following remark about the range of the datapoints.

## B.2    Definitions and notations

We start with the definition of the norm operator $\| \cdot \|$, which we use in the following two contexts.

1. If $\boldsymbol{u}$ is a $d$ dimensional vector, then $\|\boldsymbol{u}\|$ denotes the $\ell_2$ norm of $\boldsymbol{u}$, which is $\sqrt{\sum_{i=1}^{d}(\boldsymbol{u}_i)^2}$. Then $\|\boldsymbol{u} - \boldsymbol{v}\|$ is the $\ell_2$ distance between the two vectors.

2. If $M$ is a $d \times n$ matrix $M$, $\|M\|$ denotes the spectral norm of the matrix. That is,
$$\|M\| = \max_{\boldsymbol{u}, \|\boldsymbol{u}\| \leq 1} \{\|M\boldsymbol{u}\|\}$$

We follow this by defining some more structures related to random matrices.

1. For any matrix $M$, we denote $\bar{M} := \mathbb{E}[M]$. Then by definition, $\bar{X}$ is a $d \times n$ matrix such that if $i \in V_j$, the $i$-th column of $\bar{X}$ is $c_j$ (as $\mathcal{D}^{(j)}$ is a coordinate wise zero mean distribution), the ground truth center of $V_j$. Thus, we denote by $\bar{X}$ as the ground-truth or expectation matrix of $X$. Similarly $\bar{Y}$ is the center matrix of $Y$ (recall that $Y$ is the column centered matrix of $X$). Furthermore let $\bar{M}_i$ be the $i$-th column of $\bar{M}$. Then $\|\bar{y}_i - \bar{y}_{i'}\| = \|\bar{x}_i - \bar{x}_{i'}\|$ for any $(i, i')$ pair. Thus we can call $\bar{Y}$ as the ground truth matrix of $Y$.

2. Corresponding to any matrix $M$, we denote $E_M := M - \mathbb{E}[M]$.

3. **Choice of subscripts:** From hereon we use the subscript $i$ to denote the columns of $X$ and $Y$. We use the subscript $j$ for cluster identities and $\ell$ for rows of the matrices or the column vectors.

With this a background, we give a short sketch of the proof.

**Looking at the numerator and denominator separately:** Proving the relative compressibility result requires the following results in turn. Recall that the compression ratio is the ratio between pre PCA and post PCA distances between pair of datapoints and we want to lower bound "intra-community" compression ratio and upper bound "inter-community" compression ratio. This means we need the following bounds to prove Theorem B.1.

1. For the intra-community pairs of vectors, prove a lower bound on the pre PCA distance and upper bound on the post PCA distances.

2. For the inter-community pairs of vectors, prove an upper bound on the pre PCA distance and a lower bound on the post PCA distance.

We first obtain the pre PCA distance bounds, which are straightforward to obtain using the fact that the randomness in the vectors of $X$ are coordinate wise independent, and that $\|\boldsymbol{y_i} - \boldsymbol{y_{i'}}\| = \|\boldsymbol{x_i} - \boldsymbol{x_{i'}}\|$ for any $(i, i')$ pair.

### B.3 Pre PCA distances

**Lemma B.2.** *Let $\boldsymbol{y_i}$ and $\boldsymbol{y_{i'}}$ be two vectors (datapoints) of $Y$ with ground truth communities $V_j$ and $V_{j'}$ respectively. If $j = j'$ then we have $\|\boldsymbol{y_i} - \boldsymbol{y_{i'}}\| \geq \sqrt{2d\sigma_j^2 - 12\alpha\sqrt{d}\log n}$ with probability $1 - \mathcal{O}(n^{-3})$, otherwise if $j \neq j'$ then we have $\|\boldsymbol{y_i} - \boldsymbol{y_{i'}}\| \leq \sqrt{d(\sigma_j^2 + \sigma_{j'}^2) + \|c_j - c_{j'}\|^2 + 12\alpha\sqrt{d}\log n}$ with probability $1 - \mathcal{O}(n^{-3})$.*

*Proof.* We know that for any $(i, i')$ pair $\|\boldsymbol{y_i} - \boldsymbol{y_{i'}}\| = \|\boldsymbol{x_i} - \boldsymbol{x_{i'}}\|$. Using this fact we prove the bounds on the datapoints of $X$.

First we consider the case when $\boldsymbol{x_i}$ and $\boldsymbol{x_{i'}}$ belong to the same community. Then $\|\boldsymbol{x_i} - \boldsymbol{x_{i'}}\|^2 = \sum_{\ell=1}^{d}((\boldsymbol{x_i})_\ell - (\boldsymbol{x_{i'}})_\ell)^2$. Here for each $\ell$ we define $\boldsymbol{w}_\ell = (\boldsymbol{x_i})_\ell - (\boldsymbol{x_{i'}})_\ell = (c_j)_\ell + (\boldsymbol{e_i})_\ell - (c_j)_\ell - (\boldsymbol{e_{i'}})_\ell = (\boldsymbol{e_i})_\ell - (\boldsymbol{e_{i'}})_\ell$. Then $\mathbb{E}\left[\boldsymbol{w}_\ell^2\right] = \mathbb{E}[((\boldsymbol{e_i})_\ell)^2] + \mathbb{E}[((\boldsymbol{e_{i'}})_\ell)^2] = Var((\boldsymbol{e_i})_\ell) + Var((\boldsymbol{e_{i'}})_\ell)$.

We define $\sigma_{l,i}^2 = Var\left((\boldsymbol{e_i})_\ell\right)$ (to use the more familiar row major representation). Recall that both $\boldsymbol{e_i}$ and $\boldsymbol{e_{i'}}$ are sampled from $\mathcal{D}^{(j)}$ and $\sigma_j^2$ is the average of variance of the coordinates of the distribution $\mathcal{D}^{(j)}$. i.e., $\mathbb{E}\left[\sum_\ell \boldsymbol{w}_\ell^2\right] = \sum_{\ell=1}^{d} \sigma_{l,i}^2 + \sigma_{l,i'}^2 = 2d\sigma_j^2$. Now recall that the random variable $\boldsymbol{w}_\ell$ is in the range $[-2, 2]$ for any $\ell$. Then applying Hoeffding bound on this setup we get
$$\Pr\left[\sum_{\ell=1}^{d} \boldsymbol{w}_\ell^2 \leq 2d\sigma_j^2 - 12\alpha\sqrt{d}\log n\right] \leq n^{-3}.$$

Thus, if $x_i$ and $x_{i'}$ belong to the same community then with probability $1 - \mathcal{O}(n^{-3})$ we have $\|x_i - x_{i'}\| \geq \sqrt{2d\sigma_j^2 - 12\alpha^2\sqrt{d}\log n}$.

Similarly, if $x_i$ and $x_{i'}$ belong to different communities $V_j$ and $V_{j'}$, we have the random variable $w_\ell = (x_i)_\ell - (x_{i'})_\ell$ with mean $(c_j)_\ell - (c_{j'})_\ell$ (due to the difference in the centers ) and variance $\sigma_{l,i}^2 + \sigma_{l,j}^2$, where we define $c_{\ell,j} = (c_j)_\ell$. Then $\mathrm{E}\left[w_\ell^2\right] = \sigma_{l,i}^2 + \sigma_{l,j}^2 + (c_{\ell,j} - c_{\ell,j'})^2$ and

$$\mathrm{E}\left[\sum_{\ell=1}^d w_\ell^2\right] = \sum_{\ell=1}^d \sigma_{\ell,i}^2 + \sigma_{\ell,j}^2 + (c_{\ell,j} - c_{\ell,j'})^2 = d(\sigma_j^2 + \sigma_{j'}^2) + \|c_j - c_{j'}\|^2$$

Applying Hoeffding bound we get

$$\Pr\left[\sum_{\ell=1}^d w_\ell^2 \geq d(\sigma_j^2 + \sigma_{j'}^2) + \|c_j - c_{j'}\|^2 + 12\alpha\sqrt{d}\log n\right] \leq n^{-3}.$$

Thus if $x_i$ and $x_{i'}$ belong to different communities then with probability $1 - n^{-3}$ we have $\|x_i - x_{i'}\| \leq \sqrt{d(\sigma_j^2 + \sigma_{j'}^2) + \|c_j - c_{j'}\|^2 + 12\alpha\sqrt{d}\log n}$. $\qquad \square$

Now, we move into the analysis of post-PCA distances, which is the more technical part of the proof.

## B.4 Post PCA distance

**High-level idea.** The idea behind the proof is simple.

In our setup, $\bar{X} = \mathbb{E}[X]$ is the ground truth matrix, such that if the $i$-th column of $X$ belongs to $V_j$, then the $i$-th column of $\bar{X}$ is $c_j$. Thus, $\bar{X}$ is rank $k$ and thus $\|P_{\bar{X}}^k(c_j - c_{j'})\| = \|c_j - c_{j'}\|$. This implies $\bar{Y}$ has rank $k-1$ and $\|P_{\bar{Y}}^{k-1}(c_j - c_{j'})\| = \|c_j - c_{j'}\|$. The crux of the proof is to show that $P_{\bar{Y}}^{k-1}$ can be well approximated with $P_Y^{k-1}$, even when $Y$ and $\bar{Y}$ differ significantly (due to the noise).

To achieve this result we use tools from spectral analysis of random matrices, i.e. tools that study the behavior of eigenvalue and eigenvectors of random matrices. Here we face two hurdles.

1. First we note that the matrix $Y$ is rectangular and unsymmetric. The vast majority of tools in spectral analysis of random matrix theory are focused on symmetric matrices. To use this to our advantage we focus on a closely related symmetric matrix through the following symmetrization trick, which is essential to the proof. We define the matrix $Z$ as $Z := \begin{bmatrix} 0 & Y \\ Y^T & 0 \end{bmatrix}$. This is a $d+n \times d+n$ symmetric matrix. We show that $P_Y^{k-1}$ can be analyzed through $P_Z^{k-1}$ and then approximate the second projection operator using $P_{\mathbb{E}[Z]}^{k-1}$, borrowing tools from classical random matrix theory. This gives us preliminary post PCA distance bounds expressed using $\|Y - \bar{Y}\|$.

2. Then obtaining the exact bounds of Theorem B.1 require bounds on the spectral norm of $Y - \bar{Y}$. There exists a rich literature on spectral norm of random symmetric matrices with independent entries, but $Y - \bar{Y}$ does not satisfy this either. This is because, since $Y$ is obtained by subtracting the column mean from each vector of $X$, the entries of $Y$, and thus $Y - \bar{Y}$ are not independent either. To this end, we first obtain the said properties for $X - \mathbb{E}[X]$ borrowing tools from [Vu18] on our symmetrization trick, and then accommodate for the effect of centering using results from [Han14] to complete our proof.

We now describe the symmetrization trick and its implications in detail.

### B.4.1 A comparable symmetric case

We start by recalling the symmetric matrix corresponding to $Y$, $Z = \begin{bmatrix} 0 & Y \\ Y^T & 0 \end{bmatrix}$. As per our notations we denote $\bar{Z} = \mathbb{E}[Z]$ and then we have $\bar{Z} = \begin{bmatrix} 0 & \bar{Y} \\ \bar{Y}^T & 0 \end{bmatrix}$. Furthermore we have $E_Z = Z - \bar{Z}$.

Then the eigenvectors of $Z$ and singular vectors of $Y$ (and similarly $\bar{Z}$ and $\bar{Y}$) are related as follows.

*Fact* B.3. Let the left and right singular vectors of $Y$ be $\hat{l}_t, 1 \le t \le d$ and $\hat{r}_t, 1 \le t \le n$ respectively. Then the eigenvectors of $Z$ are $\frac{1}{\sqrt{2}} \begin{bmatrix} \hat{l}_t \\ \hat{r}_t \end{bmatrix}$ with eigenvalue $\hat{\lambda}_t = s_t$ and $\frac{1}{\sqrt{2}} \begin{bmatrix} \hat{l}_t \\ -\hat{r}_t \end{bmatrix}$ with eigenvalue $\hat{\lambda}_t = -s_t$ where $1 \le t \le \min(d, n)$, The same follows for $\bar{Y}$ and $\bar{Z}$.

Here we also formally define $P_M^k$ for symmetric matrices $M$ as in this case we work with eigenvectors corresponding to top eigenvalues, instead of top singular values (as in case of $Y$), for clarity.

*Remark* B.4. For any matrix $M$, we have defined $P_X^{k'}$ as the matrix whose rows are the top $k'$ singular vectors of $M$.

However, when we discuss a symmetric matrix $M'$, $P_{M'}^{k'}$ is a matrix whose rows are the eigenvectors corresponding to the top $k'$ *eigenvalues* of $M'$.

This in turn gives us the following results connecting $P_Y^{k'}$ and $P_Z^{k'}$.

*Fact* B.5. Let $0^n$ be the $n$ dimensional zero vector. Furthermore let $\boldsymbol{v}|0^n := \begin{bmatrix} \boldsymbol{v} \\ 0^n \end{bmatrix}$ for any vector $\boldsymbol{v}$. Then for any $d$-dimensional vector $\boldsymbol{v}$ we have $\|P_Y^{k'} \boldsymbol{v}\| = \sqrt{2} \|P_Z^k(\boldsymbol{v}|0^n)\|$

This result allows us to work with the symmetric matrices $Z$ and $\bar{Z}$ instead of $Y$. Now we obtain the results needed to approximate $P_Z^{k'}$ with $P_{\bar{Z}}^{k'}$.

**Difference in spectral projections of $\bar{Z}$ and $Z$:** Here we use the Davis-Kahan Theorem [DK70] along with a result by Cape et. al. [CTP19] to obtain an upper bound between the norm of difference of the leading eigenspaces of $Z$ and $\bar{Z}$ under some appropriate orthonormal rotation that we shall use to obtain our results. The main reason behind using these tools is that the SVD projection matrix due to $\bar{Z}$ is well behaved.

**Theorem B.6** (Davis-Kahan Theorem: [DK70]). *Let $D$ and $\hat{D}$ be $p \times p$ symmetric matrices, with eigenvalues $\lambda_1, \ldots, \lambda_p$ and $\hat{\lambda}_1, \ldots, \hat{\lambda}_p$ respectively. Define $E_D = \hat{D} - D$ and $\delta_{k'} = \lambda_{k'} - \lambda_{k'+1}, 1 \le k < p$. Let $U = [\boldsymbol{u_1} \ldots, \boldsymbol{u_{k'}}]$ and $\hat{U} = [\hat{\boldsymbol{u}}_1 \ldots, \hat{\boldsymbol{u}}_{k'}]$ are matrices in $\mathbb{R}^{p \times k'}$ where $\boldsymbol{u_i}$ and $\hat{\boldsymbol{u}}_i$ are eigenvectors of $D$ and $\hat{D}$ w.r.t to the $i$-th top eigenvalue. Then*

$$\left\| \sin \Theta \left( U, \hat{U} \right) \right\| \le \frac{2\|E_D\|}{\delta_{k'}} \tag{4}$$

**Theorem B.7** (Perturbation under Procrustes Transformation: [CTP19]). *Let $U$ and $\hat{U}$ be two $p \times k'$ matrices such that the columns of $U$ (and similarly $\hat{U}$) comprise of $k'$ many unit vectors that are mutually orthogonal.*

*Then there exists a $k' \times k'$ orthonormal matrix $W_U$ such that*

$$\left\| \sin \Theta \left( U, \hat{U} \right) \right\| \le \|U - \hat{U}W_U\| \le \sqrt{2} \left\| \sin \Theta \left( U, \hat{U} \right) \right\|$$

Combining them we get the following result.

**Theorem B.8.** *Given the matrices $Y$ and $\bar{Y}$ and $Z$ and $\bar{Z}$ defined as described above, there exists an orthonormal matrix $W_Z$ such that*

$$\left\| (P_Z^{k'})^T - (P_{\bar{Z}}^{k'})^T (W_Z)^T \right\| \le \frac{2\|E_Z\|}{\delta_{k'}(Y)}$$

*This in turn implies*

$$\left\| P_Z^{k'} - W_Z P_{\bar{Z}}^{k'} \right\| \le \frac{2\|E_Z\|}{\delta_{k'}(Y)}$$

Next, we obtain a result on the projection of a random vector on a $k'$ dimensional subspace.

### B.4.2 Random Projection

Now, we derive a strong bound for $\|P_M^k e\|$ where $e \in \mathbb{R}^d$ is a coordinate wise independent random vector with mean $0^d$ and $M$ is any $d \times n$ a non-negative matrix. We essentially show that for *any* $\|P_M^k e\| = \mathcal{O}(\sqrt{k})$ even though $e = \Omega(\sqrt{d})$ with high probability. Here an important condition to be satisfied is that $M$ *and* $e$ *are independent*.

Then the projection matrix $P_M^k$ is a set of $k$-orthonormal unit vectors $p_t, 1 \le t \le k$. Then the length of a projected vector $\|P_M^k e\|$ can be written down as

$$\left\|P_M^k e\right\|^2 = \sum_{t=1}^{k} \left((p_t)^T e\right)^2$$

Here one can do an entry-wise analysis of the terms $\left((p_t)^T e\right)^2$ but that forces a bound of the form that $\|P_M^k e\| \le \sqrt{k}(\sigma + \sqrt{16 \log(nk)})$ with probability $1 - \mathcal{O}(n^{-3})$. Instead, we recall a result from the Vu [VW15].

**Lemma B.9** ([VW15]). *There are constants $C_0, C_1$ such that the following happens. Let $e$ be a random vector in $\mathbb{R}^d$ such that its coordinates are independent random variables with $0$ mean and variance $\sigma^2$. Assume furthermore that the coordinates are bounded by $\alpha$ in their absolute value. Let $H$ be a subspace of dimension $k$ and let $\Pi_H(e)$ be the length of the orthogonal projection of $e$ onto $H$. Then for any $n$ we have*

$$\Pr\left(\Pi_H(e) \ge \sigma\sqrt{k} + C_1\alpha\sqrt{\log n}\right) \le n^{-3}$$

Now, let us consider the case where $H$ is the subspace covered by the top $k$ many orthonormal eigenvectors of $M$, denoted as $p_t, 1 \le t \le k$. Then the projection of $e$ onto $H$ can be written as $\sum_{t=1}^{k} \langle p_t, e \rangle p_t$. Then we have $(\Pi_H(e))^2 = \sum_{t=1}^{k} \langle p_t, e \rangle^2 = \sum_{t=1}^{k} \left((p_t)^T e\right)^2$. This is exactly $\|P_M^k(e)\|^2$. Summarizing, we get the following result with respect to the matrix $\bar{Z}$.

**Corollary B.10.** *Let $P_{\bar{Z}}^{k'}$ be as defined above. Let $e$ be a $d$-dimensional random vector with each entry having zero mean and variance at most $\sigma^2$. Then with probability $1 - n^{-3}$ we have,*

$$\|P_{\bar{Z}}^k(e)\| \le \sigma\sqrt{k} + C_1 \cdot \alpha \cdot \sqrt{\log n}$$

We are now in a position to obtain our preliminary pots PCA distance bounds when the projection dimension is $k - 1$.

### B.4.3 Preliminary post PCA bounds

**Preliminary intra-community bounds:** We start by obtaining the post PCA distance $\|\Pi_Y^{k-1}(y_i - y_{i'})\|$ where both $y_i$ and $y_{i'}$ belong to the same community $V_j$.

**Lemma B.11.** *Let $y_i$ and $y_{i'}$ be two columns of the data matrix $Y$ belonging to the same community $V_j$. Then for some constants $C_1$ we have*

$$\|P_Y^{k-1}(y_i - y_{i'})\| \le 2\sqrt{2}\frac{\|Z - \bar{Z}\| \cdot \|y_i - y_{i'}\|}{\delta_{k-1}(Y)} + 2\sqrt{2}\left(\sigma\sqrt{k} + C_1 \cdot \alpha \cdot \sqrt{\log n}\right) \quad (5)$$

*with probability $1 - \mathcal{O}(n^{-3})$.*

*Proof.* Initially we have $\|P_Y^{k-1}(y_i - y_{i'})\| = \sqrt{2}\|P_Z^{k-1}(y_i|0^n - y_{i'}|0^n)\|$. Here we use the facts that the spectral projection operators due to $Z$ and $\bar{Z}$ are close up to some orthonormal rotation and that $\bar{Z}$ and $y_i - y_{i'}$ are independent. Furthermore $y_i - y_{i'} = c_j + e_i - c_X - c_{j'} - e_{i'} + c_X = e_i - e_{i'}$, where $c_X$ is the centering vector which is a zero mean random vector.

Then we have for any $k - 1$ dimensional orthonormal matrix $W$,

$$\|P_Z^{k-1}(y_i|0^n - y_{i'}|0^n)\| \le \|(P_Z^{k-1} - WP_{\bar{Z}}^{k-1})(y_i|0^n - y_{i'}|0^n)\| + \|WP_{\bar{Z}}^{k-1}(y_i|0^n - y_{i'}|0^n)\|$$

$$\le \|P_Z^{k-1} - WP_{\bar{Z}}^{k-1}\| \cdot \|y_i|0^n - y_{i'}|0^n\| + \|WP_{\bar{Z}}^{k-1}(e_i|0^n - e_{i'}|0^n)\|$$

$$\le \|P_Z^{k-1} - WP_{\bar{Z}}^{k-1}\| \cdot \|y_i|0^n - y_{i'}|0^n\| + \|WP_{\bar{Z}}^{k-1}e_i|0^n\| + \|WP_{\bar{Z}}^{k-1}e_{i'}|0^n\|$$

From Theorem B.8 we have that for a choice of $W$ $\|P_Z^{k-1} - WP_{\bar{Z}}^{k-1}\| \leq \frac{2\|Z-\bar{Z}\|}{\delta_{k-1}(Z)} = \frac{2\|Z-\bar{Z}\|}{\delta_{k-1}(Y)}$.

Next, we can analyze $\|WP_{\bar{Z}}^{k-1}\boldsymbol{e_i}|0^n\| + \|WP_{\bar{Z}}^{k-1}\boldsymbol{e_{i'}}|0^n\|$ as $\bar{Z}$ and the vectors are independent of each other. Then applying Corollary B.10 with probability $1 - \mathcal{O}(n^{-3})$ we have $\|WP_{\bar{Z}}^{k-1}\boldsymbol{e_i}|0^n\| + \|WP_{\bar{Z}}^{k-1}\boldsymbol{e_{i'}}|0^n\| \leq 2\sigma\sqrt{k-1} + 2C_1\sqrt{\log n}$. This completes the proof.

$\square$

**Preliminary inter-community bounds.**    Now, we move to the inter-community results. In this part $P_{\bar{Z}}^{k-1}$ plays an important role. This is because as per our discussion $\|P_{\bar{Y}}^{k-1}(\mathbf{c_j} - \mathbf{c_{j'}})\| = \|\mathbf{c_j} - \mathbf{c_{j'}}\|$. This implies

$$\|P_{\bar{Z}}^{k-1}(\bar{Y}_i|0^d - \bar{Y}_{i'}|0^d)\| = \|\mathbf{c_j} - \mathbf{c_{j'}}\| \tag{6}$$

Using this result we then prove the following inter-community post PCA bound.

**Lemma B.12.** *Let $\boldsymbol{y_i}, \boldsymbol{y_{i'}}$ be two columns of the data matrix $Y$ so that $i \in V_j$ and $i' \in V_{j'}$, where $j \neq j'$. Then for the constant $C_1$ we have*

$$\|P_Y^{k-1}(\boldsymbol{y_i} - \boldsymbol{y_{i'}})\| \geq \sqrt{2}\left(\|\mathbf{c_j} - \mathbf{c_{j'}}\| - 2\left(\sigma\sqrt{k} + C_1 \cdot \alpha \cdot \sqrt{\log n}\right) - \frac{2\|Z-\bar{Z}\| \cdot \|\boldsymbol{y_i} - \boldsymbol{y_{i'}}\|}{\delta_{k-1}(Y)}\right) \tag{7}$$

*with probability $1 - \mathcal{O}(n^{-3})$.*

*Proof.* As before we have $\|P_Y^{k-1}(\boldsymbol{y_i} - \boldsymbol{y_{i'}})\| = \sqrt{2}\|P_Z^{k-1}(\boldsymbol{y_i}|0^n - \boldsymbol{y_{i'}}|0^n)\|$. Then we proceed with a basic decomposition. We have for any $k-1$ dimensional matrix $W$,

$$\|P_Z^{k-1}(\boldsymbol{y_i}|0^n - \boldsymbol{y_{i'}}|0^n)\|$$
$$\geq \|WP_{\bar{Z}}^{k-1}(\mathbf{c_j}|0^n - \mathbf{c_{j'}}|0^n)\| - \|WP_{\bar{Z}}^{k-1}(\boldsymbol{e_i}|0^n - \boldsymbol{e_{i'}}|0^n)\| - \|(P_Z^{k-1} - WP_{\bar{Z}}^{k-1})(\boldsymbol{y_i}|0^n - \boldsymbol{y_{i'}}|0^n)\|$$

Now, we have $\|WP_{\bar{Z}}^{k-1}(\mathbf{c_j}|0^n - \mathbf{c_{j'}}|0^n)\| = \|P_{\bar{Y}}^{k-1}(\mathbf{c_j} - \mathbf{c_{j'}})\| = \|c_j - c_{j'}\|$.

Next from Lemma B.11 we have $\|WP_{\bar{Z}}^{k-1}(\boldsymbol{e_i}|0^n - \boldsymbol{e_{i'}}|0^n)\| \leq 2\left(\sigma\sqrt{k} + C_1\sqrt{\log n}\right)$ with probability $1 - \mathcal{O}(n^{-3})$.

Finally from Lemma B.11 we know we can upper bound $\|(P_Z^{k-1} - WP_{\bar{Z}}^{k-1})(\boldsymbol{y_i}|0^n - \boldsymbol{y_{i'}}|0^n)\|$ with $\frac{2\|Z-\bar{Z}\|}{\delta_{k-1}(Y)} \cdot \|\boldsymbol{y_i} - \boldsymbol{y_{i'}}\|$, which completes the proof.

$\square$

At this point, we have obtained the pairwise post-PCA intra-community and inter-community distance bounds in terms of $\|\boldsymbol{y_i} - \boldsymbol{y_{i'}}\|, \|Z - \bar{Z}\|, k, \sigma$ and $\delta_{k-1}(Y)$. Here $\delta_{k-1}(Y)$ is the spectral gap of $Y$ and we already have bounds on $\|\boldsymbol{y_i} - \boldsymbol{y_{i'}}\|$. Next, we obtain bounds on $\|Z - \bar{Z}\|$ and then put together the results obtained so far to prove Theorem B.1.

### B.4.4   Spectral norm of the square marrix

First, we note down a result by Vu [Vu18] for upper bounds on the spectral norm of random matrices with independent entries.

**Theorem B.13** (Norm of random symmetric matrix [Vu18]). *Let $E$ be a $n \times n$ random symmetric matrix where each entry in the upper diagonal is an independent random variable with $0$ mean and $\sigma$ variance, then there is a constant $C_0$ such that*

$$\Pr\left[\|E\| \geq C_0\sigma\sqrt{n}\right] \leq n^{-3}$$

*where $\sigma^2 \geq C_1\frac{\log n}{n}$.*

However, since the entries of $Y$ are not independent, the same follows with $E_Z$. To bypass this issue we define the matrix $B := \begin{bmatrix} 0 & X \\ X^T & 0 \end{bmatrix}$ and then $\bar{B} := \mathbb{E}[B] = \begin{bmatrix} 0 & \bar{X} \\ \bar{X}^T & 0 \end{bmatrix}$

Furthermore recall that $E_M = M - \mathbb{E}[M]$. Then we have the following results.

1. $\|E_Z\|$ is the largest eigenvalue of $E_Z$, which is same as the largest singular value of $E_Y$, that we denote as $s_1(E_Y)$.

2. $\|E_B\|$ is same as the largest singular value of $E_X$, that we denote as $s_1(E_X)$.

Furthermore we have from Theorem B.13 that $\|E_B\| \leq C_0\sigma\sqrt{n}$ with probability $1 - \mathcal{O}(n^{-3})$. Finally we connect $s_1(E_Y)$ with $s_1(E_X)$. To do so, note that $E_Y$ is the centered matrix of $E_X$. This follows from the fact that $E_Y = Y - \bar{Y}$ and $E_X = X - \bar{X}$. Then we use the following result by Hanoine [Han14].

**Theorem B.14** ([Han14]). *Let $M$ be a rank $m$ matrix and $\bar{M}$ be the matrix obtained upon centering, with singular values (in descending order) $s_1, \ldots, s_m$ and $\bar{s}_1, \ldots, \bar{s}'_{m-1}$ respectively. Then for any $1 \leq i < m$ we have $s_i \geq s'_i \geq s_{i+1}$.*

Using this result we get
$$\|E_Z\| = s_1(E_Y) \leq s_1(E_X) \leq \|E_B\|$$

Now, we bound $\|E_B\|$, i.e. $\left\| \begin{bmatrix} 0 & E_X \\ (E_X)^T & 0 \end{bmatrix} \right\|$. This is a $(d+n) \times (d+n)$ random symmetric matrix with zero mean and maximum variance $\sigma^2$. Then applying Theorem B.13 we get the following bound.

**Lemma B.15.** *Recall that we define $\mathcal{N} = C_0\sigma\sqrt{d+n}$. Then in the setting of Lemma B.11 we have $\|Z - \bar{Z}\| \leq \mathcal{N}$ with probability $1 - \mathcal{O}(n^{-3})$*

Against this backdrop we summarize our bounds to prove Theorem B.1.

### B.5 Proof of Theorem B.1

From Lemma B.2 we have the lower bound on the intra-community distances and upper bound on the inter-community distances. Similarly, we can also use the results to obtain lower bound for the intra-community case. It is easy to see that if (i,i') belong to the same community $V_j$ then with probability $1 - \mathcal{O}(n^{-3})$, $\|\boldsymbol{y_i} - \boldsymbol{y_i}\| \leq \sqrt{2d\sigma_j^2 + 12\alpha\sqrt{d}\log n}$.

Substituting this and the bound on $\|Z - \bar{Z}\|$ to Lemma B.11 we have with probability $1 - \mathcal{O}(n^{-3})$

$$\|P_Y^{k-1}(\boldsymbol{y_i} - \boldsymbol{y_{i'}})\| \leq 2\sqrt{2}\left( \sigma\sqrt{k} + C_1 \cdot \alpha \cdot \sqrt{\log n} + \frac{\mathcal{N} \cdot \sqrt{2d\sigma_j^2 + 12\alpha\sqrt{d}\log n}}{\delta_{k-1}(Y)} \right)$$

Similarly for the inter-community with $i \in V_j, i' \in V_{j'}$ from Lemma B.12 we have

$$\|P_Y^{k-1}(\boldsymbol{y_i} - \boldsymbol{y_{i'}})\| \geq \sqrt{2}\left( \|\mathbf{c_j} - \mathbf{c_{j'}}\| - 2\left(\sigma\sqrt{k} + C_1 \cdot \alpha \cdot \sqrt{\log n}\right) - \frac{\mathcal{N} \cdot \sqrt{\|\mathbf{c_j} - \mathbf{c_{j'}}\|^2 + d(\sigma_j^2 + \sigma_{j'}^2) + 12\alpha\sqrt{d}\log n}}{\delta_{k-1}(Y)} \right)$$

Finally, using the bounds of Lemma B.2 and applying a union bound on the total $n^2$ pairs of datapoints completes the proof of the theorem.

## C  Spatially unique centers and proof for the outlier detection theorem

The primary quantity that is hard to interpret in a dataset with an underlying community structure is $\delta_{k-1}(Y)$. Here we make some observations. First note that $\delta_{k-1}(Y) = s_{k-1}(Y) - s_k(Y)$. Now, $s_{k-1}(Y) \geq s_k(X)$ and $s_k(Y) \leq C\sigma\sqrt{d+n}$ where the latter term comes from the fact that $s_k(Y) \leq \|E_Y\| \leq \|E_X\| \leq C\sigma\sqrt{d+n}$. This follows from a simple application of Weyl's inequality and the effect of centering on eigenvalues. For simplicity, we consider the case when $s_k(X) \geq 4C\sigma\sqrt{d+n}$. We will come back to this and show that this assumption does make sense. Then, we have $\delta_{k-1}(Y) \geq 0.66 s_k(X)$. Next note that $s_k(X) \geq s_k(\mathbb{E}[X]) - \|E\|$.

This then implies that given the aforementioned conditions, we have
$$\delta_{k-1} \geq 0.25 s_k(\mathbb{E}[X]) \tag{8}$$
where $\mathbb{E}[X]$ is the center matrix, where each column is the center of the community the corresponding point belongs to.

**Bounds on the singular values of the center matrix for $\gamma$-spatially unique centers**   Here, we make a connection between $s_k(\mathbb{E}[X])$ and the notion of spatially unique centers.

Given a $n_1 \times n_2$ matrix $M$, we define the minimum hyperplane distance, $\mathsf{dist}_M$ as

$$\mathsf{dist}_M = \min_j \min_{\mathbf{v} \in \mathsf{Span}(M_{-j})} \|M_j - \mathbf{v}\|$$

where $M_j$ represents the $j$-th column of $M$ and $M_{-j}$ denotes the set of all columns of $M$ except the $j$-th one. That is, it denotes the minimum distance between a data point and the span of the rest of the data points.

We have the following classic result of matrix theory.

**Lemma C.1** ([RV08]). *For any $n_1 \times n_2$ matrix $M$, the smallest singular value $s_{\min}(M)$ is lower bounded by $\frac{1}{\sqrt{n_2}} \cdot \mathsf{dist}_M$.*

Now, this result does not directly help us as $\mathbb{E}[X]$ has multiple identical columns (it is after all a $d \times n$ rank $k$ matrix) and we only get a lower bound of 0. However, we can do a simple two-step analysis to get something nicer.

Consider the matrix $\hat{C}$ which contains $k$ columns that are each copy of one of the centers of $X$. Then from the definition of $\gamma$-spatially unique centers, we immediately have the following.

*Fact* C.2. If $X$ comes from a setup with $\gamma$-spatially unique centers then $s_k(\hat{C}) = s_{\min}(\hat{C}) \geq \frac{\gamma}{\sqrt{k}}$.

Next, let the size of the underlying communities in $X$. Then we know that $\mathbb{E}[C]$ has at least $\min_j |V_j|$ many copies of $\hat{C}$ in it (along with other columns corresponding to the larger communities). That means that the singular values in $\mathbb{E}[X]$ is at least $\sqrt{\min_j |V_j|}$ times the singular values in $\hat{C}$. This gives us the following result.

**Lemma C.3.** *Let $X$ be generated from $\gamma$-spatially unique centers and let the minimum size of the underlying communities be $\Omega(n/k)$. Furthermore, assume $s_k(X) \gg \|E_X\|$. Then we get $\delta_{k-1}(Y) \geq \frac{C \cdot \gamma \sqrt{n}}{k}$ for some constant $C$.*

*Proof.* This simply comes from putting the bounds on $\hat{C}$ and multiplying them with $\sqrt{\min_j |V_j|}$ and then connecting it with Equation 8. $\qquad \square$

Now, to go back to the assumption of $s_k(X) \geq 4C\sigma\sqrt{d+n}$, consider that $n = \Omega(d)$ (this is where will work from hereon), then the assumption holds as long as $\gamma \geq \sigma k$. Now, once we have this result, we can then obtain our main Theorem C.4 in the setting of Spatially unique centers.

**Theorem C.4** (Relative compression with spatially unique centers). *Let $X$ be a $d \times n$ dataset $k$ many $\gamma$-spatially unique centers where the size of the smallest community is $\Omega(n/k)$. Then there is a constant $C_1$ such that for all intra-community pairs in $V_j$, the compression ratio is upper-bounded as*

$$\Delta_{X,k-1}(i,i') \geq \frac{\sigma_j \sqrt{d}}{C_1 \left( \sigma\sqrt{k} + \alpha\sqrt{\log n} + \frac{2\sigma \cdot \sigma_j \cdot k\sqrt{d}}{\gamma} \right)} \tag{9}$$

*Similarly for any $i \in V_j$ and $i' \in V_{j'}$, the inter-community compression ratio is upper-bounded as*

$$\Delta_{X,k-1}(i,i') \leq$$
$$\frac{\sqrt{(\sigma_j^2 + \sigma_{j'}^2)d^2 + \|\mathbf{c_j} - \mathbf{c_{j'}}\|^2}}{C_1 \left( \|\mathbf{c_j} - \mathbf{c_{j'}}\| - 2\sigma\sqrt{k} - \alpha\sqrt{\log n} - \frac{\sigma \cdot \sqrt{\sigma_j^2 + \sigma_{j'}^2} \cdot k \cdot \sqrt{d}}{\gamma} \right)} \tag{10}$$

*with probability $1 - \mathcal{O}(1/n)$.*

*Proof.* For simplicity of the statements we have made several assumptions, most of them to consider the harder setting (heavy noise). That is, $\sigma_j^2 d = \Omega(\max_{j'} \|\mathbf{c_j} - \mathbf{c_{j'}}\|)$. Furthermore, we assume $\sigma$ is sufficiently large so that $\sigma^2 d \geq 100\alpha\sqrt{d}\log n$ (this happens as long as $\alpha = o(d^{1/4})$). This implies that the pre-PCA intra and inter-community distances are $\Theta(2\sigma_j\sqrt{d})$ and $\Theta\left(\sqrt{\sigma_j^2 + \sigma_{j'}^2}\sqrt{d}\right)$ respectively.

Next, in the intra-community compression ratio bound we have the term

$$\frac{2\mathcal{N} \cdot \sqrt{\sigma_j^2 d + 12\sqrt{d}\log n}}{\delta_{k-1}(Y)} = \frac{2\sigma\sqrt{d+n} \cdot \sqrt{\sigma_j^2 d + 12\alpha\sqrt{d}\log n}}{0.25\gamma\sqrt{n}/k}$$

Here recall that we assume $n = \Omega(d)$ which implies $2\sigma\sqrt{d+n} \leq C\sigma\sqrt{n}$ for large enough $n$. Furthermore, $12\alpha\sqrt{d}\log n$ is dominated by $\sigma_j^2 d$. Combining we get

$$\frac{2\sigma\sqrt{d+n} \cdot \sqrt{\sigma_j^2 d + 12\alpha\sqrt{d}\log n}}{0.25\gamma\sqrt{n}/k} = \frac{C\sigma\sqrt{n}\sigma_j\sqrt{d} \cdot k}{0.25\gamma\sqrt{n}} = \frac{8C\sigma\sigma_j \cdot k \cdot \sqrt{d}}{\gamma}$$

Similarly the bound

$$\frac{2\sigma\sqrt{d+n} \cdot \sqrt{\|\mathbf{c_j} - \mathbf{c_{j'}}\|^2 + 2(\sigma_j^2 + \sigma_{j'}^2)d + 12\alpha\sqrt{d}\log n}}{\delta_{k-1}(Y)}$$

can be simplified to $\frac{C\sqrt{\sigma_j^2 + \sigma_{j'}^2}k\sqrt{d}}{\gamma}$

Combining these bounds directly gets us result. □

Then, Theorem 2.5 is immediately implied, as follows.

## C.1 Proof of Theorem 2.5

We know that $\gamma \geq C\max\{\sigma\sqrt{k}d^{1/4}, \sigma\sqrt{k} + \alpha\log n\}$. Furthermore, we assume $\max_{i,j}\|\mathbf{c_i} - \mathbf{c_j}\| \ll \sigma\sqrt{\mathbf{d}}$, which is the heavy noise setting. The other case follows the same way. Let $C > 100C_1$.

Furthermore, note that $\|\mathbf{c_i} - \mathbf{c_j}\| \geq \gamma$.

Then we have

$$\frac{2\sigma \cdot \sigma_j \cdot k\sqrt{d}}{\gamma} \leq 0.01\sigma_j\sqrt{k}d^{1/4}$$

Similarly, we have

$$\frac{\sqrt{\sigma_j^2 + \sigma_{j'}^2} \cdot k \cdot \sqrt{d}}{\gamma} \leq 0.01\sigma\sqrt{k}d^{1/4}$$

Then, the denominator of the lower bound on the intra-community compression ratio is upper bounded by $0.02\sigma\sqrt{k}d^{1/4}$, and the denominator of the lower bound on the inter-community compression ratio is lower bounded by $\|\mathbf{c_i} - \|\mathbf{c_j}\| - 0.02\sigma\sqrt{k}d^{1/4} \geq 0.98\sigma\sqrt{k}d^{1/4}$.

Then, the intra-community compression ratio is lower bounded by $10\sqrt{k}\sqrt{d^{1/4}}$ and the inter-community compression ratio is upper bounded by $0.05\sqrt{k}\sqrt{d^{1/4}}$, obtaining the separation described in the Theorem 2.5.

## C.2 Proofs for variance of compression ratios

Having discussed the compression ratio bounds in the context of $\gamma$-spatially unique centers, we continue with the theoretical support for our outlier detection method in the random-mixture-outlier model. We recall the definition of this model for ease of exposition.

**Definition C.5** (Mixture model with outliers (revisited)). Let $X$ be a $d \times n$ dataset with the partition $V_1, \ldots V_k, \hat{V}$, a set of $k$ centers $\{\mathbf{c_j}\}_{j=1}^{k}$ and distributions $\{\mathcal{D}^{(j)}\}_{j=1}^{k} + 1$ with the following generation method.

1. *clean points:* If $i \in V_j, 1 \leq j \leq k$, $\boldsymbol{x_i} = \mathbf{c_j} + \boldsymbol{e_i}$ where $\boldsymbol{e_i}$ is sampled from $\mathcal{D}^{(j)}$.

2. *outliers:* If $i \in \hat{V}$, then we sample $p_{i,1}, \ldots p_{i,k} \in [0.5, 1]$. Then $\boldsymbol{u_i} = \sum_j \alpha_{i,j} \mathbf{c_j} + \boldsymbol{e_i}$ where $\alpha_{i,j} = \frac{p_{i,j}}{\sum_j p_{i,j}}$ and $\boldsymbol{e_i}$ is sampled from $\mathcal{D}^{(k+1)}$.

Let $|\hat{V}| = n_o$ and $n = n_o + n_c$. To keep the results simple, we make the average variance of each distribution $\mathcal{D}^{(j)}$ same, which is $\sigma'$.

The concept of Algorithm 1 is simple. If each cluster has a large number of points, then even if there are a large number of outliers generated from the random-mixture-outlier model, the outliers will have a lower variance of compression than all the clean points.

First, let us obtain a lower bound on the variance of the compression ratio of clean points under the conditions of Theorem 2.8. We know that any clean point has high intra-community compression ratios. This implies that the expectation of the compression ratio for this point is high. On the other hand, the inter-compression ratio values are low. So just calculating the variance on the inter-community points yields a large value.

For the sake of simplicity, we will define $\gamma \geq 2\beta\sqrt{\sigma}kd^{1/4}$. Then if we can show that under the other settings of Theorem 2.8, there is a separation in the variance of the compression ratios of the clean points and the outliers whenever $\beta \geq C'\frac{\sigma\sqrt{\log n}}{\sigma'}$, we prove the theorem.

**Lemma C.6.** *Let there be $n_c$ clean points in the random-mixture-outlier setting where $\min_j |V_j| = \Omega(n/k)$ and $\gamma \geq 2\beta\sqrt{\sigma}kd^{1/4}$. Then the variance of all such points are lower bounded as $C_4 \cdot \frac{n_c - |V_j|}{n} \cdot \frac{d^{1/4}}{k} \cdot \left(\frac{\beta\sigma'}{\sigma} - \frac{\sigma}{\beta \cdot \sigma'}\right)$ with probability $1 - \mathcal{O}(1/n)$.*

*Proof.* Consider any point $\boldsymbol{x_i} \in V_j$. Then the inter-compression ratio of $\boldsymbol{x_i}$ with any intra-community point is lower bounded by $\frac{0.25\sigma'\sqrt{d}}{\sigma\sqrt{k}+\alpha\sqrt{\log n}+2\sqrt{\sigma\sigma'}d^{1/4}/(\beta)} \geq \frac{0.25\beta\sigma'}{\sigma}d^{1/4}$ with probability $1 - \mathcal{O}(1/n)$.

Then the average of the compression ratios for $\boldsymbol{x_i}$ is lower bounded as $\frac{0.25\sigma'/\sigma d^{1/4}|V_j|}{n} \geq \frac{C_3\beta\sigma'/\sigma d^{1/4}}{k}$

On the other hand, probability $1 - \mathcal{O}(1/n)$ we have that for any inter-community point, the compression ratio is upper-bounded with $\frac{2\sigma\sqrt{d}}{(\gamma-(2\sigma\sqrt{k}-\alpha\sqrt{\log n}-C_2\sigma'd^{1/4})} \leq \frac{2\sigma\sqrt{d}}{\beta k\sigma'd^{1/4}/C_2} \leq \frac{2C_2\sigma/\sigma'd^{1/4}}{\beta \cdot k}$.

Then, the variance of compression of $\boldsymbol{x_i}$ is lower bounded by

$$C_4 \cdot \frac{n - |V_j|}{n} \cdot \left(\frac{d^{1/4}}{k} \cdot \left(\frac{\beta\sigma'}{\sigma} - \frac{\sigma}{\beta\sigma'}\right)\right)^2$$

$\square$

Now, we aim to upper-bound the variance of compression for outliers. Here we want to show that since the underlying signal in any outlier is apart from the signal of any other point, they generally have a lower compression ratio with any other point, which then implies a lower variance of compression ratio.

First, we show that as long as there are not too many outliers, their underlying centers (which are random mixtures of the community centers) will not be too close (which implies they will not have a high compression ratio).

**Lemma C.7** (Distance between signals of the outliers). *Let there be $n_o$ many outliers in the dataset generated via the random-mixture model where $k \geq \log n$. Let the set of outliers be $\hat{V}$. Then, for with probability $1 - \mathcal{O}(n)$, $\min_{\boldsymbol{u},\boldsymbol{v}\in\hat{V}} \|\boldsymbol{u} - \boldsymbol{v}\| \geq \frac{\gamma}{\log n}$.*

*Proof.* Let $|\hat{V}| = n_o$. Then, for the underlying mixture-center any two points, denoted as $\boldsymbol{u} = \sum_j \alpha_{1,j} \mathbf{c_j}$ and $\sum_j \alpha_{2,j} \mathbf{c_j}$ we say they are $\epsilon$-far if $\min_j |\alpha_{1,j} - \alpha_{2,j}| \geq \epsilon$.

Now, note that for any $\epsilon$-far mixture-centers, we have $\|\boldsymbol{u} - \boldsymbol{v}\| \geq 0.5\epsilon\gamma$.

Now, it is easy to see that the probability that there is a pair of mixed centers that is not $\epsilon$-far is $n_0^2 \cdot (\epsilon)^k$. Then, setting $\epsilon = 1/\log n$ and applying $k \geq \log n$ gives that even for $n_0 = n/2$, there all pairs of mixture centers are $1/\log n$-far with probability $1 - \mathcal{O}(1/n)$. $\qquad\square$

Then, we show that in such a case, the variance of compression for any outlier point is quite low even when measured crudely.

**Lemma C.8** (Variance of compression of outliers). *Let there be a set of $n_o$ many outliers so that the underlying mixture-centers are pairwise $1/\log n$-far Then under the condition of Lemma C.6, we have that the variance of compression for any outlier is upper bounded by $\left(\frac{4C_2 \log n(\sigma/\sigma')d^{1/4}}{\beta \cdot k}\right)^2$ with probability $1 - \mathcal{O}(1/n)$.*

*Proof.* Consider any outlier $\boldsymbol{u_i} \in V_0$. First, consider the compression ratio between $\boldsymbol{u_i}$ and any $\boldsymbol{v}$ that is clean. Where $\boldsymbol{u_i} = \sum_j \alpha_j \mathbf{c_j} + \boldsymbol{e_{i'}}$ and $\boldsymbol{v} = \mathbf{c_{j'}} + \boldsymbol{e_i}$.

Next, remember that as every $\alpha_j \geq 1/2k$, we have $\max_j \alpha_j \leq 0.5$.

Then, from the definition of $\gamma$-spatially unique centers we have

$$\|\sum_j \alpha_j \mathbf{c_j} - \mathbf{c_{j'}}\| \geq \|\sum_{j \neq j'} \alpha_j \mathbf{c_j} - (1 - \alpha_{j'})\mathbf{c_{j'}}\| \geq 0.5\|\sum_{j \neq j'} \alpha_j \mathbf{c_j} - \mathbf{c_{j'}}\| \geq 0.5\gamma$$

Then, following the analysis of Lemma C.6, we can show that in all such cases, the compression ratio is upper bounded by $\frac{4C_2\sigma/\sigma'd^{1/4}}{\beta \cdot k}$.

On the other hand, consider any two outliers. Then their compression ratios are upper bounded by $\frac{4C_2 \log n\sigma/\sigma'd^{1/4}}{\beta \cdot k}$ (essentially replacing $\gamma$ by $\gamma/\log n$ in the center-distance calculation).

Then, we can upper bound the variance of compression for an outlier as

$$\frac{1}{n}\left(|\hat{V}|(\frac{4C_2 \log n\sigma/\sigma'd^{1/4}}{\beta \cdot k})^2 + (n - |\hat{V}|)(\frac{4C_2\sigma/\sigma'd^{1/4}}{\beta \cdot k})^2\right) \leq \left(\frac{4C_2 \log n(\sigma/\sigma')d^{1/4}}{\beta \cdot k}\right)^2 \text{[applying } k \geq \log n\text{]}$$

$\qquad\square$

**Proof of Theorem 2.8** Lemma C.6 shows that in the setting of Theorem 2.8, the variance of compression ratios for a clean point is lower bounded by $C_4 \cdot \frac{n-|V_j|}{n} \cdot \left(\frac{d^{1/4}}{k} \cdot \left(\frac{\beta\sigma'}{\sigma} - \frac{\sigma}{\beta\sigma'}\right)\right)^2$.

Next, Lemma C.8 shows that the variance of compression ratios for an outlier is upper-bounded as $\left(\frac{4C_2 \log n(\sigma/\sigma')d^{1/4}}{\beta \cdot k}\right)^2$ Both the aforementioned happen for all outlier and clean points with probability $1 - \mathcal{O}(1/n)$.

Then, to show that with high probability, the variance of compression ratios of any clean point is higher than the variance of compression ratios of any outlier is

$$C_4 \cdot \frac{n-|V_j|}{n} \cdot \left(\frac{d^{1/4}}{k} \cdot \left(\frac{\beta\sigma'}{\sigma} - \frac{\sigma}{\beta\sigma'}\right)\right)^2 > \left(\frac{4C_2 \log n(\sigma/\sigma')d^{1/4}}{\beta \cdot k}\right)^2$$

$$\implies \frac{\sqrt{d}}{k^2} \cdot \left(\frac{n-|\hat{V}|}{n} \cdot \frac{\beta\sigma'}{\sigma} - \frac{C_5 \log n\sigma}{\sigma'\beta}\right) > 0 \qquad\qquad \text{[For some constant } C_5\text{]}$$

$$\implies \frac{\sqrt{d}}{k^2} \cdot \left(\frac{0.5\beta\sigma'}{\sigma} - \frac{C_5 \log n\sigma}{\sigma'\beta}\right) > 0 \qquad\qquad \text{[As } n - |\hat{V}| \geq 0.5n\text{]}$$

Then, as long as $\beta \geq 10C_5\sigma/\sigma'\sqrt{\log n}$, this equation is satisfied.

## D   Projection with more principal components

Here we show some results in the case of $k' = k - 1 + c$. The main challenge in theoretically proving our bounds for $k' \neq k - 1$ comes from Theorem B.8. A key ingredient towards proving Theorem B.1 is the following spectral gap.

$$\left\| (P_Z^{k'})^T - (P_{\bar{Z}}^{k'})^T W \right\| \leq \frac{2\|E_Z\|}{\delta_{k'}(Y)}$$

In general we work with the natural assumption $\delta_{k-1}(Y) >> \|E_Z\|$. However, in our model we have $s_k(Y) = \mathcal{O}(\|E_Z\|)$. This follows from Weyl's inequality, which states that if $Z = \bar{Z} + E_Z$ and $(k')$-th singular value of $B$ is 0, then $(k - 1 + c)$-th singular value of $Z$ is upper bounded by $\mathcal{O}(\|E_Z\|)$ for any $c > 0$.

Thus $\delta_{k'}(Y) = \mathcal{O}(\|E_Z\|)$ for any $k' \geq k$, and our previous results alone cannot prove relative compressibility.

Here we bypass this issue to a loose but non-trivial extent. First we note that the inter-community compression can only decrease if the the projection dimension increases. Thus we have that for any $k' \geq k$, $\Delta_{X,k'}(i, i') \leq \Delta_{X,k-1}(i, i')$.

**Theorem D.1.** *Let us consider the random vector model as in Theorem B.1. Let Let $k' = k - 1 + c$ and any $0 < f < 1$. Then we have that with probability $1 - \mathcal{O}(1/n)$,*

1. *If $(i, i')$ is an inter-community pair, then $\Delta_{k',X}(i, i') \leq \Delta_{k-1,X}(i, i')$*

2. *If $(i, i')$ is an intra-community pair, then*

$$\Delta_{k-1,Y}(i, i') \geq \frac{\sqrt{\|\mathbf{c_j} - \mathbf{c_{j'}}\|^2 + d(\sigma_j^2 + \sigma_{j'}^2) + 12\sqrt{d}\log n}}{\sqrt{\left\| P_Y^{k-1}(\boldsymbol{y_i} - \boldsymbol{y_{i'}}) \right\|^2 + 4C_0^2 \sigma^2(d+n)c^2 f^2}}$$

*for all but $c^2/f^4$ pairs of points with probability $1 - \mathcal{O}(1/n)$.*

*Proof.* The inter-community bound follows from definition and the numerator of the intra-community bound follows from Lemma B.2. We now prove the denominator (post PCA distance bounds) for the intra-community case.

Let us denote with $P_Y^{k_1,k_2}$ the projection operator due to the $k_1$-th to $k_2$-th top singular vectors of $Y$. Then for any vector $\boldsymbol{u}$ we have $\|\Pi_X^{k'}(\boldsymbol{u})\| = \sqrt{\|\Pi_X^{k-1}(\boldsymbol{u})\|^2 + \|P_Y^{k,k'}(\boldsymbol{u})\|^2}$.

Then we are left with bounding $\|P_Y^{k,k'}(\boldsymbol{u})\|^2$ where $\boldsymbol{u} = \boldsymbol{y_i} - \boldsymbol{y_{i'}}$ so that $i \in V_j, i' \in V_{j'}$. We aim to show that if $k' - k$ is small then this value is small as well.

We first represent $Y$ with its SVD decomposition. $\boldsymbol{l_\ell}$ and $\boldsymbol{r_\ell}$ represent the $\ell$-th left singular vector and right singular vector of $Y$ respectively. Then we have $Y = \sum_{\ell=1}^{t} s_i(Y)\boldsymbol{l_\ell}(\boldsymbol{r_\ell})^T$ where $t = rank(Y)$. Then the projection of $\boldsymbol{y_i}$ due to the $\ell$-th principal component of $X$ is $\langle \boldsymbol{l_\ell}, \boldsymbol{y_i} \rangle = s_\ell(Y)r_{\ell,i}$ where $r_{\ell,i}$ is the $i$-th entry of the $\ell$-th right singular vector. Then we have

$$\leq s_k(Y)\sqrt{\sum_{\ell=k}^{k'}(r_{\ell,i})^2}$$

Here recall that each $\boldsymbol{r_\ell}$ is a $n$-dimensional vector with unit norm, i.e. $\|\boldsymbol{r_\ell}\| = 1$. Then for any $f < 1$, the number of coordinates of $\boldsymbol{r_\ell}$ that are larger than $f$ is less than $1/f^2$. Thus considering all the $k \leq \ell \leq k - 1 + c$, the total number of entries that are larger than $f$ is less than $c/f^2$. Then, for all but $c/f^2$ many points $\boldsymbol{y_i}$ we have $\|P_Y^{k,k-1}(\boldsymbol{y_i})\| \leq s_k(Y) \cdot f \cdot c$. Here we substitute $s_k(Y) \leq \|E_Y\| \leq C_0\sigma\sqrt{d+n}$ with probability $1 - \mathcal{O}(n^{-3})$.

This implies that with probability $1 - \mathcal{O}(1/n)$ $\|P_Y^{k,k-1}(\boldsymbol{y_i} - \boldsymbol{y_{i'}})\| \leq 2C_0\sigma\sqrt{d+n}cf$ for all but $c^2/f^4$ pairs of points. This concludes our proof.

$\square$

It should be noted that this is a much looser bound as compared to our $(k-1)$-PC compression metric, especially in the paradigm where noise dominates the ground truth distances. As we discussed, the main reason that Theorem B.1 does not directly work for $k' > k - 1$ can be pinned down to the following technical challenge.

### D.0.1 Technical challenges in understanding PCA

The technical challenge is getting a better upper-bound on $\|(P_Z^{k-1} - P_{\bar{Z}}^{k-1})(e)\|$ than $\|P_Z^{k-1} - P_{\bar{Z}}^{k-1}\| \cdot \|e\|$ for a random vector $e$. In fact, $\|(P_Z^{k-1} - P_{\bar{Z}}^{k-1})e\|$ equals $|(P_Z^{k-1} - P_{\bar{Z}}^{k-1})\| \cdot \|e\|$ only if $e/\|e\|$ is a unit vector along which $(P_Z^{k-1} - P_{\bar{Z}}^{k-1})$ realizes its spectral norm, which is unlikely to be the case for most noise $e$ vectors, due to the inherent randomness in them. A better analysis of this term will allow us to extend the result of Theorem B.1 beyond $k' = k - 1$, which is what we observe in reality. For real datasets, the compression factor does not change much if the PCA dimension is changed by a small value. Furthermore, a tighter understanding of $\|(P_Z^{k-1} - P_{\bar{Z}}^{k-1})e\|$ will also enable to us make progress towards proving optimality of perhaps the simplest spectral clustering algorithm for the SBM problem, as conjectured by [Vu18]. There has indeed been some progress very recently [MZ24, MZ23] in some very specific settings, i.e. SBM (stochastic block model). Generalizing these results to the random vector mixture model is an outstanding open question.

## E  Experiments

### E.1  Summary of datasets

First, we present a summary of the datasets.

| Dataset | # of clusters | # of cells | # of genes (features) |
|---|---|---|---|
| Koh | 9 | 531 | 48,981 |
| Kumar | 3 | 246 | 45,159 |
| Simkumar4easy | 4 | 500 | 43,606 |
| Simkumar4hard | 4 | 499 | 43,638 |
| Simkumar8hard | 8 | 499 | 43,601 |
| Trapnell | 3 | 222 | 41,111 |
| Zheng4eq | 4 | 3,994 | 15,568 |
| Zheng4uneq | 4 | 6,498 | 16,443 |
| Zheng8eq | 8 | 3,994 | 15,716 |

Table 4: Summary of data

### E.2  Community-wise average compression ratios

In Section 4 we showed the average of community-wise average of intra and inter-community compression ratios for the datasets in [DRS20] for PCA dimension=$k - 1$. Here we present the results for each community of the datasets. We observe that even in the community-level metric, the intra-community compression ratio is higher than the inter-community compression ratio for all datasets.

### E.3  NMI and purity index improvement for PCA-dim=$k - 1$

Now, we continue with providing more experimental results. First, we note down the NMI improvement when $5\%$ and $10\%$ of the points are removed in the setting of PCA dimension= $k - 1$.

Next, we add the initial purity scores when running PCA( dimension=$k-1$)+K-means on the datasets in Table 6.

Then the improvement in purity index due to $5\%$ and $10\%$ points removal are recorded in Figures 5 and 6.

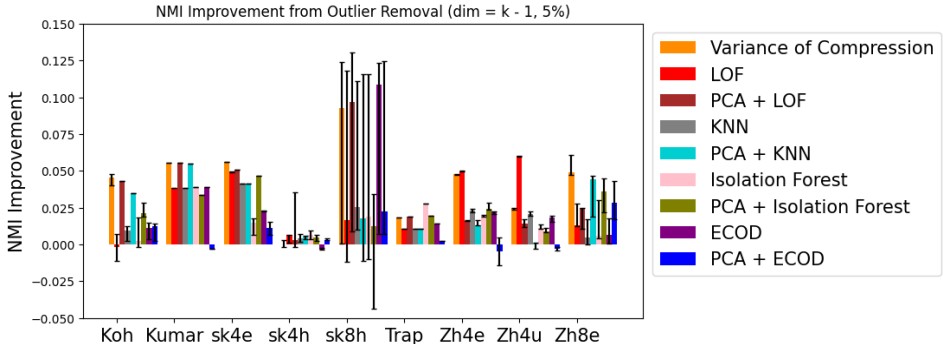

Figure 3: NMI improvement via removing $5\%$ points

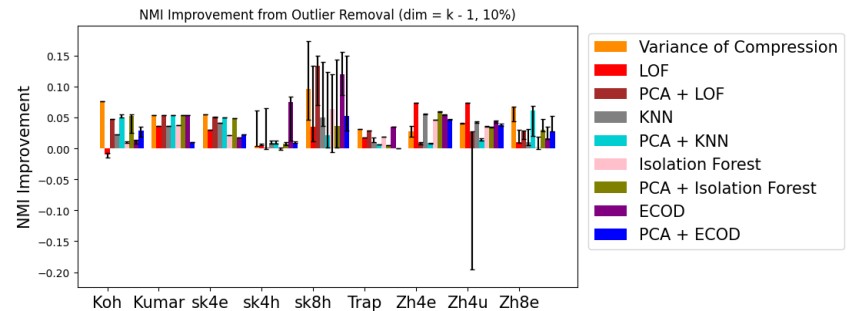

Figure 4: NMI improvement via removing $10\%$ points

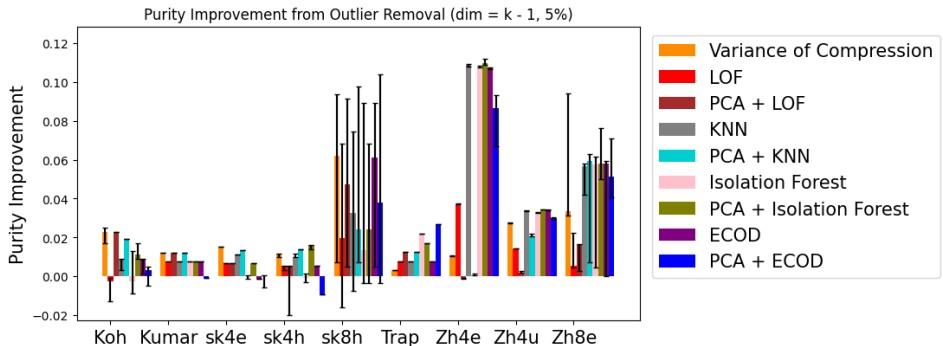

Figure 5: Purity score improvement via $5\%$ outlier removal

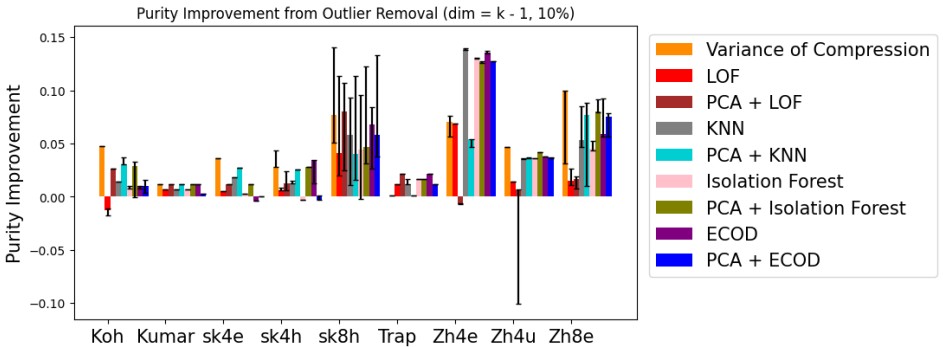

Figure 6: Purity score improvement via $10\%$ outlier removal

|  | Community wise average compression ratio | | | | | | | | |
|---|---|---|---|---|---|---|---|---|---|
| Koh (Inter) | 2.417 | 2.530 | 2.714 | 2.523 | 2.649 | 2.948 | 2.352 | 2.696 | 2.018 |
| Koh (Intra) | 7.678 | 9.829 | 6.966 | 6.041 | 6.757 | 8.424 | 6.686 | 7.382 | 7.463 |
| Kumar (Inter) | 2.107 | 2.105 | 1.696 | - | - | - | - | - | - |
| Kumar (Intra) | 15.969 | 13.577 | 14.889 | - | - | - | - | - | - |
| Simkumar4easy (Inter) | 4.534 | 3.724 | 3.200 | 2.850 | - | - | - | - | - |
| Simkumar4easy (Intra) | 15.673 | 17.083 | 15.554 | 14.924 | - | - | - | - | - |
| Simkumar4hard (Inter) | 5.984 | 5.653 | 4.960 | 4.472 | - | - | - | - | - |
| Simkumar4hard (Intra) | 15.173 | 16.722 | 14.500 | 13.807 | - | - | - | - | - |
| Simkumar8hard (Inter) | 4.425 | 4.668 | 4.397 | 5.233 | 4.390 | 4.004 | 3.998 | 3.681 | - |
| Simkumar8hard (Intra) | 9.177 | 10.571 | 8.785 | 8.639 | 9.390 | 9.526 | 8.699 | 10.172 | - |
| Trapnell (Inter) | 4.491 | 7.401 | 7.228 | - | - | - | - | - | - |
| Trapnell (Intra) | 9.202 | 10.248 | 10.122 | - | - | - | - | - | - |
| Zheng4eq (Inter) | 2.117 | 1.762 | 2.828 | 2.889 | - | - | - | - | - |
| Zheng4eq (Intra) | 6.135 | 6.250 | 7.947 | 6.223 | - | - | - | - | - |
| Zheng4uneq (Inter) | 2.059 | 1.753 | 2.870 | 2.176 | - | - | - | - | - |
| Zheng4uneq (Intra) | 5.839 | 6.351 | 7.335 | 5.514 | - | - | - | - | - |
| Zheng8eq (Inter) | 1.981 | 2.922 | 1.655 | 1.936 | 2.567 | 2.594 | 2.802 | 2.726 | - |
| Zheng8eq (Intra) | 4.306 | 4.533 | 4.540 | 4.997 | 4.254 | 5.598 | 5.244 | 4.300 | - |

Table 5: Community-wise Inter and Intra-Community Compression Ratios

| Dataset | Purity of PCA + k-means |
|---|---|
| Koh | 0.895 |
| Kumar | 0.983 |
| Simkumar4easy | 0.918 |
| Simkumar4hard | 0.563 |
| Simkumar8hard | 0.667 |
| Trapnell | 0.604 |
| Zheng4eq | 0.715 |
| Zheng4uneq | 0.873 |
| Zheng8eq | 0.568 |

Table 6: Purity index before data removal (PCA dim = $k - 1$)

## E.4 Different PCA dimension choice

Finally, we show that our experiments on real-world data, both for average compression as well as clustering accuracy improvement through outlier detection, are fairly stable to a change in the PCA dimension. The average compression ratios can be found in Table 7. The NMI and purity index baselines can be found in Tables 8 and 9 respectively.

| Dataset | Avg. intercluster compression | Avg. intracluster compression |
|---|---|---|
| Koh | 2.246 | 4.484 |
| Kumar | 1.742 | 5.576 |
| Simkumar4easy | 3.007 | 6.496 |
| Simkumar4hard | 4.161 | 6.461 |
| Simkumar8hard | 3.537 | 4.948 |
| Trapnell | 3.259 | 4.204 |
| Zheng4eq | 1.969 | 4.246 |
| Zheng4uneq | 1.893 | 4.081 |
| Zheng8eq | 2.139 | 3.491 |

Table 7: Relative compression on RNA-seq datasets when PCA dimension is $2k$

| Dataset | NMI of PCA + k-means |
|---|---|
| Koh | 0.861 |
| Kumar | 0.924 |
| Simkumar4easy | 0.744 |
| Simkumar4hard | 0.235 |
| Simkumar8hard | 0.440 |
| Trapnell | 0.293 |
| Zheng4eq | 0.710 |
| Zheng4uneq | 0.724 |
| Zheng8eq | 0.560 |

Table 8: NMI before data removal (PCA dim $= 2k$)

| Dataset | Purity of PCA + k-means |
|---|---|
| Koh | 0.898 |
| Kumar | 0.984 |
| Simkumar4easy | 0.910 |
| Simkumar4hard | 0.561 |
| Simkumar8hard | 0.658 |
| Trapnell | 0.608 |
| Zheng4eq | 0.720 |
| Zheng4uneq | 0.878 |
| Zheng8eq | 0.574 |

Table 9: Purity score before data removal (PCA dim $= 2k$)

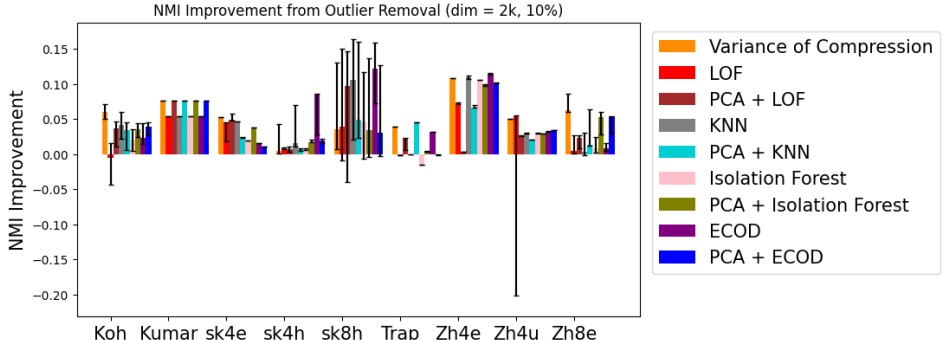

Figure 7: NMI improvement via removing $10\%$ points when PCA dimension is $2k$

For brevity, we show the improvement in NMI and purity index for $10\%$ point removal in Figures 7 and 8 respectively. As one can observe, our method continues to be the most consistent, being the best method in most datasets. Indeed, in this case our performance is even comparatively better than in the case of PCA-dimension=$k-1$.

## F Future directions

In this paper, we have quantified PCA's denoising effect in high dimensional noisy data with underlying community structure via the metric of compression ratio. As an application, we have designed an outlier detection method that improves the community structure of datasets. We note two interesting theoretical and algorithmic questions.

i) Providing a more tight bound on the compression ratio seems an exciting and hard direction.

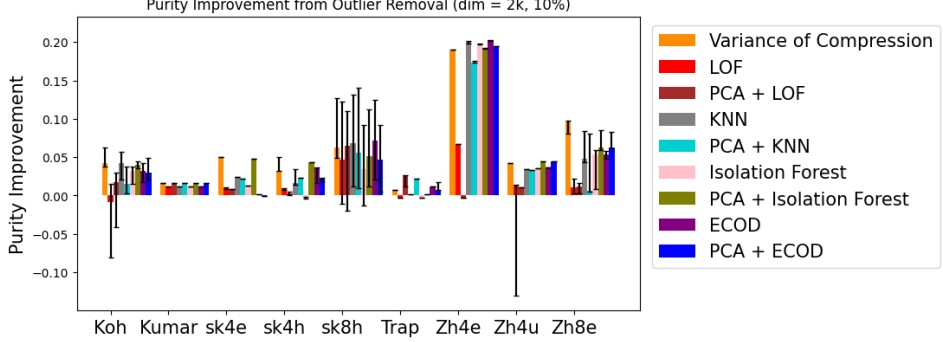

Figure 8: Purity score improvement via removing $10\%$ points when PCA dimension is $2k$

ii) Using compression ratio as a metric for clustering algorithms also seems an interesting direction, especially for single-cell-RNA-seq datasets.

