# OpenReview forum: "Capturing the denoising effect of PCA via compression ratio"
_NeurIPS.cc/2024/Conference — NeurIPS 2024 poster_

### Official Review · Reviewer_QWbP · 2024-07-15

**Soundness:** 3
**Presentation:** 3
**Contribution:** 3
**Rating:** 5
**Confidence:** 3

**Summary:**

In this paper, the authors propose a novel metric called compression ratio to capture the effect of PCA on denoising which can significantly reduce the distance of data points belonging to the same community while reducing inter-community distance relatively mildly. They try to explain this phenomenon through both theoretical proofs and experiments on real-world data. In addition, they design a straightforward algorithm that could be used to detect outliers and provide many experimental results to demonstrate its superiority over existing methods.

**Strengths:**

1. the paper studies an important question that how to characterize the improvement of dimension reduction via PCA in denoising and clustering.
2. the proof part is solid and sound
3. the experiment part is rich and convincing, validating the theoretical analysis part

**Weaknesses:**

The paper only conduct experiments on single-cell RNA-seq, which is not convincing. Results on more datasets are expected.

**Questions:**

Can more metric comparison be included such ARI?
Table 3 the middle 2 columns (Purity) they are the same, I guess one is 5% and another one should be 10% instead of 5%?

---

> ### Author Rebuttal · Authors · 2024-08-07
>
> We thank the reviewer for their valuable comments. Please find our response below.
>
> **Regarding only using single-cell data:**
>
> We first note that single-cell is indeed a very important data-type with various applications in immunology, neuroscience, and others. The Science Journal placed it as the breakthrough of the year in 2017. Also, many machine learning papers use single-cell data primarily for experiments, including recent NeurIPS papers [1,2]
>
> Additionally, we have tested that the compression ratio phenomenon of PCA is widely present in **high dimensional noisy data**. For example, we ran our experiments on a certain AFLP dataset [1]. Furthermore, this phenomenon can also be observed in popular image datasets such as MNIST and F-MNIST if they are artificially corrupted with high variance noise. Although we observed the compression ratio gap on these datasets, they do not have natural outliers. Hence, we did not include the compressibility results due to the space constraints of NeurIPS papers.
>
> By contrast, single-cell data is a very good fit for high-dimensional noisy data with natural outliers. Some erroneous cells are the results of mixtures of different cell types or cells disproportionately affected by noise, which makes this datatype an ideal candidate for our outlier detection method.
>
> Overall, we are also interested in testing more high-dimensional noisy datasets with natural outliers. If the reviewer has such datasets in mind, please let us know, and we will be happy to test more.
>
> **Other clustering metric:**
> We can surely show improvements in other clustering metrics, such as the ARI. Following the reviewer’s recommendation, we ran these experiments and observed that our method provides the second-best rank for ARI. Here, we want to note that ARI seems not to give an accurate picture of the clustering outcome if the clusters have varied sizes, and indeed, all of the outlier detection methods had weaker improvements in the ARI metric compared to both NMI and purity.
>
> As a summarizing step, we calculated the ranks across different metrics (NMI, ARI, and purity), and our method (variance of compression) had the best overall rank by a significant margin. For example, for dimension k-1, we had a rank of 2.8, whereas the next best rank was 3.7.
>
> On a separate note, we will fix the heading of the table as pointed out by the reviewer (yes, the right column is for 10% purity).
>
> [1] Gong, Jing, et al. "xTrimoGene: an efficient and scalable representation learner for single-cell RNA-seq data." NeurIPS2023.
>
> [2] Palma, Alessandro, et al. "Modelling single-cell RNA-seq trajectories on a flat statistical manifold." NeurIPS 2023 AI for Science Workshop. 2023.
>
> ------
>
> We hope this answers the reviewer's questions, and we will be happy to answer any other queries they have.

---

### Official Review · Reviewer_mA6h · 2024-07-17

**Soundness:** 3
**Presentation:** 3
**Contribution:** 2
**Rating:** 5
**Confidence:** 4

**Summary:**

This paper introduces compression ratio, defined as the ratio of pre-and-post-PCA distances for a pair of observations, for outlier detection tasks. The authors demonstrated that this metric could capture the effect of PCA on high-dimensional data with moderate noise and proposes that points with lower variance of compression ratio and do not share a common signal with others are more likely to be outliers. The proposal was validated in both simulations and on real-world scRNA-seq datasets.

**Strengths:**

1. The authors provided a detailed statistical setup for the problem and provided theoretical guarantees of components of the algorithm.
2. The idea is quite simple and elegant
3. The simulation shows that with moderate noise, the proposed method achieves good outlier removal methods.

**Weaknesses:**

1. More robustness/sensitivity analysis would be needed to help understand how widely applicable this simple ratio is.
2. The current real data application is not well-motivated: while identifying potentially mislabeled cell types are interesting, these are often better treated as deconvolution and/or mixture model problems.

**Questions:**

1. In Figure 1, why is the variance of compression achieving moderate to low ranks?
2. The simulated outliers seem to come from a convex combination of existing clusters — have the authors considered more general outlier cases?
3. I would encourage the authors to provide more simulation results with higher dimensionality (right now it uses d=1,000 < n=3,000)?
4. On the selection of dimension for PCA, it looks like choosing anywhere from k to 2k (where k is the true number of “community”) is not too bad — in practice is there any advice and guidance on choosing k in the first place that performs reasonably?
5. From Algorithm 1, it seems like returning these indices requires at least doing O(n^2) distance after performing PCA — adding the theoretical and empirical time complexities on some of these datasets would benefit this method's users.
6. I would encourage the authors to compare the results on benchmark datasets used for outlier removal, especially since that is the primary application highlighted.

**Limitations:**

Detailed in questions.

---

> ### Author Rebuttal · Authors · 2024-08-07
>
> We thank the reviewer for their valuable comments. Please find our response below.
>
>
> **Motivation behind real-world application:**
> We thank the reviewer for their input. First, we observe that such mixture model approaches are predominantly used in data with access to bulk-RNA seq data, which can be less complex than single-cell RNA seq data [1,2].
>
> More importantly, we note that the highly cited work [3]  laid down a standard pipeline for single-cell RNA seq data. Here, preprocessing is followed by clustering, and then the clustering is used to find differentially expressed genes (DEG) in the data (i.e., features that have predominantly high values in one cluster compared to other clusters). In such cases, a better separation of the underlying data can be beneficial in better DEG detection. As our outlier removal improves the performance of clustering algorithms such as PCA+K-Means, we hope it may be of interest and use to bioinformaticians.
>
> **Answer to questions**
>
> **Weaker performance in Figure (1):**
> We start by noting an observation from the outlier detection survey paper we referenced (HHH+22) that the success of an outlier detection algorithm depends on how closely the assumption behind the algorithm matches that of the dataset. In our paper, we theoretically studied our outlier detection algorithm in the case where outliers are generated as random mixtures of community centers further perturbed by noise, which is essentially the same as random mixtures of the points in the communities.
>
> Figure 1(b) is for a *different* kind of outliers. Here, outlier points are generated by adding a “higher variance noise” to the corresponding community center compared to normal points for that community. Note that this is an outlier model different from the one we theoretically study. In fact, outlier detection methods like LOF are known to have a very good performance in such settings (HHH+22). We wanted to showcase that our method can be competitive even when the outlier model differs from our theoretical model. We hope this will further increase confidence in our outlier detection method.
>
> **Types of outliers:** As we have noted above, apart from the outlier model where outliers are convex combinations of points in the communities (which we study theoretically), we also provide results for the case where outliers are points with higher variance noise (which we present in Figure 1(b)). Apart from that, we also test our method on a large set of single-cell data.
>
> **Simulations for d>n:**  Following the recommendation by the reviewer, we also ran experiments with d>n by setting d=4000 and n=2000. We obtained identical performance as the ones shown in the paper. In fact, for the higher noise case, our outlier detection performance was comparatively even better than the other methods. We shall add this result to the paper.
>
> **Choice of PCA-projection dimension:** We did not consider this question in this paper. This is indeed an important question. In our experience, simple techniques such as elbow plots on the eigenvalue of the covariance matrix seem to be a good technique for obtaining a good choice. Additionally, there exists a large body of work on choosing the correct PCA dimension for noisy high-dimensional data.
>
> **Run-time complexity:** Indeed, calculation of all O(n^2) distances is necessary for obtaining the variance, bringing the time complexity to O(n^2*d).
> For the datasets we consider, our algorithm runs under 3 minutes, with it running under 5 seconds for the smaller datasets (less than 1000 points). We also want to point out that the distance calculations are highly parallelizable.
>
> Furthermore, theoretically we can improve this runtime significantly in our model. For each point, we may sub-sample a few points O(sqrt(n) log n) and then calculate the compression ratio with these points to obtain the variance of compression. This would reduce the run-time to only consider n^{1.5} log n distances. Empirical verification of such speedups is a future direction.
>
>
> [1] Song, Liyang, et al. "Mixed model-based deconvolution of cell-state abundances (MeDuSA) along a one-dimensional trajectory." Nature Computational Science 3.7 (2023): 630-643
>
> [2] Chu, Tinyi, et al. "Cell type and gene expression deconvolution with BayesPrism enables Bayesian integrative analysis across bulk and single-cell RNA sequencing in oncology." Nature Cancer 3.4 (2022): 505-517
>
> [3] Heumos, Lukas, et al. "Best practices for single-cell analysis across modalities." Nature Reviews Genetics 24.8 (2023): 550-572.
>
>
> -----
>
> We hope this answers the reviewer's questions, and we will be happy to answer any other queries they have.

---

### Official Review · Reviewer_1iTr · 2024-07-19

**Soundness:** 3
**Presentation:** 3
**Contribution:** 3
**Rating:** 6
**Confidence:** 3

**Summary:**

This paper studies the denoising effect of PCA  using a novel metric called "compression ratio". The metric is defined as the ratio of the pre and post-PCA between two points. The authors note that when the dataset has a community structure, outlier points tend to have a flatter distribution of compression ratios w.r.t other data points, whereas inlier points have a larger compression ration w.r.t to other intra-community data points. This insight is used to design a simple outlier-detection algorithm for data with community structure. Under certain assumptions on the data distribution, the authors show that this outlier identification algorithm succeeds with a constant probability. The authors also provide many experimental results on both synthetic and real-world datasets to show that their proposed methods outperform existing algorithms.

**Strengths:**

- The paper studies a problem of good interest and provides a clean, well-motivated solution.
- The theoretical results are interesting

**Weaknesses:**

- Because of the theoretical nature of the paper, there are many assumptions such as knowing the number of outliers, and assumptions on the noise distribution. These assumptions do not always hold in real-world data.

**Questions:**

- In Figure 1, it seems like the proposed method (Variance of compression) performs worse than most other methods. Can the authors provide more information regarding Figure 1b?
-

**Limitations:**

See questions and weaknesses.

---

> ### Author Rebuttal · Authors · 2024-08-07
>
> We thank the reviewer for their valuable comments. Please find our answers below.
>
> **Regarding the theoretical assumptions:**
> Our theoretical setting is a generalization of several popular unsupervised models, such as the Gaussian mixture models and the stochastic block model. The noise distribution we consider is SubGaussian with very high variance. Such distributions capture noise observed in many real-world scenarios, such as datasets with bounded entries (as all bounded variables are subGaussian).
>
> Next, we note that our outlier detection algorithm *does not* need knowledge of the number of outliers. We prove that if the data has outliers in the random mixture model setting, these outliers will have the lowest variance of compression.
>
>
> **Regarding the weaker performance in figure 1(b):**
> We start by noting an observation from the outlier detection survey paper we referenced (HHH+22) that the success of an outlier detection algorithm depends on how closely the assumption behind the algorithm matches that of the dataset. In our paper, we theoretically studied our outlier detection algorithm in the case where outliers are generated as random mixtures of community centers further perturbed by noise, which is essentially the same as a random mixture of the points in the communities.
>
> Figure 1(b) is for a *different* kind of outliers. Here, outlier points are generated by adding a “higher variance noise” to the corresponding community center compared to normal points for that community. Note that this is an outlier model different from the one we theoretically study. In fact, outlier detection methods like LOF are known to have a very good performance in such settings (HHH+22). We wanted to showcase that our method can be competitive even when the outlier model differs from our theoretical model. We hope this will further increase confidence in our outlier detection method.
>
>
>
> ------
>
> We hope this answers the reviewer's questions, and we will be happy to answer any other queries they have.

---

### Official Review · Reviewer_51iT · 2024-07-23

**Soundness:** 2
**Presentation:** 2
**Contribution:** 2
**Rating:** 3
**Confidence:** 4

**Summary:**

The paper proposes a new measure called ‘compression ratio’ to determine how effectively PCA compresses data. For subspace clustered data the authors show that if the signal directions for each cluster are well-separated, the compression ratio is larger for within clusters than compared to between clusters. The paper also proposes a method for outlier detection through this compression ratio.

**Strengths:**

The paper theoretically shows that for clustered data that are well separated and each cluster centroid is situated in a subspace, not representable by the span of centroids, the compression within clusters is greater than compressions between clusters, implying the formation of tight clusters in the reduced low-dimensional representation.

**Weaknesses:**

1. The assumptions in the paper are quite strong in my opinion. Especially the assumption of nearly orthonormal cluster centroids is unrealistic for low-dimensional data ($n \gg d$) where PCA is usually applied.

2. The outlier detection algorithm requires a run of the PCA to find the compression ratio. However, it is well known that PCA is not at all outlier robust. In that case how are the compression ratios reliable? The theoretical guarantees in Theorem 2.8 does not seem to match this intuition. Is this guarantee only an artefact of the restrictive assumption on the subGaussianity of the outliers?

**Questions:**

1. Usually, it is common practice to calculate the information lost in PCA based on the ratio of the sum of the eigenvalues. How does the compression ratio relate to the variance explained? A clarification regarding this is warranted.
2. How do the authors manage to run vanilla PCA on single-cell RNAseq data on which $d \gg n$? Do they use a different version of the PCA?
3. How does the method compare to other robust PCA techniques?  What are the advantages of their proposal compared PCA-based custering methods such as IF-HCT-PCA?

Jiashun Jin. Wanjie Wang. "Influential features PCA for high dimensional clustering." Ann. Statist. 44 (6) 2323 - 2359, December 2016. https://doi.org/10.1214/15-AOS1423

Also see my questions in the weakness section.

**Limitations:**

Yes, limitations are discussed.

---

> ### Author Rebuttal · Authors · 2024-08-07
>
> We thank the reviewer for their review. Please find our responses below.
>
> **Comments regarding weaknesses:**
>
> **Regarding unrealistic conditions of centers:**
> The reviewer comments that the assumption of centers being nearly orthonormal is unrealistic for the case of n>>d (where n is the number of data points and d is the dimension). This is **an incorrect evaluation of our theoretical contribution** in our understanding. First, we point out that our assumption can be interpreted as each center (there are $k$ centers in total)  being not *a linear combination of others*. This assumption is *much weaker* than nearly orthonormal. Our assumption covers many natural benchmark models, such as the stochastic block model. Secondly, even the realizability of this weaker condition has no connection with the comparative values of n and d.
>
>
> **Outlier detection in PCA:**
> As the reviewer pointed out, PCA is not robust to outliers in  *all* scenarios. In this paper, we mainly focus on *high-dimensional noisy data*, which we present in the random mixture vector model with subGaussian noise. First, we observe the compression ratio phenomenon of PCA in simulation and *all* the datasets that we consider. Then, we use this phenomenon to detect outliers in this model. Furthermore, The overall performance of our outlier detection algorithm on a large set of real-world datasets is a strong support of its efficacy.
>
>
> **Answer to questions:**
>
> Q1) Regarding “explained variance”: In this paper, the compression ratio is defined on k-dimensional PCA projection and the amount of information retained is dependent on factors such as k and the variance of the noise in each community. Among our datasets, the top k singular values range from 3%-10% of the sum of the singular values, depending on the noise level of the datasets. There is a very weak correlation between this value and compressibility phenomena.
>
> Q2) The vanilla PCA algorithm works for **any** rectangular matrices, and as such, we do not need to use a different method for d>>n and n>>d.
>
> Q3) The goal of the paper is to analyze the “compressibility” of PCA itself, and as such, we do not focus on other PCA variants. It is an interesting future direction to see if similar compressibility phenomena are observed in other variants of PCA, and we thank the reviewer for their suggestion.
> *Regarding the PCA based clustering method:* Please note that we do not propose a clustering algorithm in this paper. Rather, we use the compression ratio metric to observe the denoising effect of PCA in noisy high-dimensional data with underlying community structure. Next, we use this metric to design an *outlier detection* algorithm. Therefore, comparisons with PCA-based clustering methods are not relevant to the paper.
>
>
>
>
> ------
>
> We hope this answers the reviewer's questions, and we will happily answer any other queries they have.

---

> > ### Comment · Reviewer_51iT · 2024-08-12
> > **Response to Authors**
> >
> > Thank you for the detailed rebuttal. However, I still have some reservations regarding the assumptions as they are quite strong and does not hold for practical senarios. For example the assumption won't hold for $k > d$. Additionally, if I understand correctly, the subgaussianity of the outliers appears to be crucial for the theoretical guarantees, potentially excluding cases with unboundedly large outliers. This undermines the purpose of developing of being outlier-robust. Please correct me if I’m mistaken.
> >
> > Given that the paper also proposes an outlier-robust PCA and evaluates its effectiveness through clustering performance, I believe it should be compared with similar recent methods, such as IF-HCT-PCA and other robust PCA techniques. Unfortunately, the current experimental setup does not seem to address such comparisons.
> >
> > Thus, I am inclined to keep my scores as is.

---

> ### Author Response · Authors · 2024-08-12
>
> As we mentioned in our paper (even abstract, first sentence of the second paragraph), our main focus in this paper is high-dimensional noisy data. Here $k$ is the number of clusters and $d$ is the dimension of vectors. In single-cell data, $k$ is usually smaller than $50$ and $d$ is usually larger than $20000$. For another example, in popular image data sets such as MNIST and F-MNIST, $k=10$ and $d=784$. The condition $d\gg k$  is also true for many many other domains.
>
> In the case of $k\geq d$, we  believe PCA or spectral algorithms are not the correct approaches. For example, many spectral algorithms (including PCA) projected the high-dimensional data into $\Theta(k)$ dimensional vectors in applications to data/models with $k$-clusters. If $k\geq d$, we believe the projection will remove some useful information. See the link for spectral clustering https://en.wikipedia.org/wiki/Spectral_clustering .
>
> If the reviewer insisted that our algorithm failed in low-dimensional data, then we agree with the reviewer and have nothing to discuss with, since $k\geq d$ is not in our consideration.
>
> Also, in the entirety of our paper, we only say that our algorithm is robust under the choice of the projected dimension, and it is very different from the concept of robust PCA. Using the compression ratio of PCA as an outlier detection algorithm has no connection to robust PCA. We are not sure why the reviewer connected our result with robust PCA. Please see the link of robust PCA (https://en.wikipedia.org/wiki/Robust_principal_component_analysis).

---

### Comment · Area_Chair_MRB4 · 2024-08-07
**rebuttal?**

Dear authors (and reviewers),

This paper has split reviews, all close to the border between acceptance and rejection. A lot of NeurIPS submissions fall into this category, and if there's no strong rebuttal, usually these papers end up being rejected.

So I highly suggest the authors write a strong rebuttal to each of the reviews. (If you don't think this is possible, you can choose to withdraw the paper). If you think the reviewers were not clear enough in their criticism, you can always ask them to elaborate.

Reviewers, please read each others' reviews (and the rebuttals when written). As a reminder to everyone, the OpenReview platform allows us to have a dialogue, so you can post small questions/comments and responses at any time.

Best, Area Chair

---

### Decision · Program_Chairs · 2024-09-25

**Decision:**

Accept (poster)

**Comment:**

The paper had 3 weakly positive reviews, and 1 negative review. The negative review seems most influenced by the fact that the method doesn't work well in low dimensions, but the authors point out they explicitly target the high-dimensional case and provide clear examples of high-dimensional data.  Ignoring the lowest review (as an "outlier"), the paper's average score is still just below the typical cutoffs for a target 20% to 25% NeurIPS acceptance rate. This is just to say that the reviewing process found this paper borderline. Unfortunately the reviewers were not too active in discussion after the rebuttal period.

Hence, the AC read the paper to form their own opinion. At a high-level, this simply seems like a good paper. It has theory and experiments. The theory makes assumptions, as most theory does, but the authors are pretty good about discussing these assumptions and limitations, and include artificial simulations to test the method under different levels of noise-to-separation. Ultimately the real-world experiments, comparing with a very reasonable set of alternative methods, show that the method does well in the type of biological data problem considered, which somewhat validates the assumptions.

I don't see any major flaws, and I like the contributions: some theory, and a new practical method (very simple, and the only parameter is the number of outliers). This reminds me of k-means (ie., Lloyds algorithm) itself: there have been hundreds of papers that propose methods to improve on k-means, but everyone still uses vanilla k-means because it's simple, not-bad, and time-tested. So there's a lot to be said for simple methods, especially when there is some kind of theory behind it.

Thus I'm happy to recommend acceptance.

I do have a few minor comments which I encourage the authors to address, in addition to any remaining issues raised by reviewers.
-  "… K-Means, which is also known as spectral clustering."  The structure of the sentence made this vague, but if you are saying that reducing dimension with PCA and then doing k-means is the same as spectral clustering, I disagree. For spectral clustering, k-means is done on the eigenvectors themselves, not after applying the eigenvectors to the data. Furthermore, spectral clustering has an explicit metric or graph structure leading to an explicit graph Laplacian, which is then used for the eigendecomposition. For PCA, it can be thought of as either doing the SVD on the data, or the eigendecomposition of the Gram matrix. But even if you think of the Gram matrix as the graph Laplacian, I think it's still distinct from the usage you have in mind.   I think I'm right, but regardless of whether I am right or wrong about this, this connection needs more explanation.
- I assume that these new outlier methods outcompete old standards, such as DBSCAN (which clusters and finds outliers at the same time), right?
- Line 150 mentions $d \ge 10 \alpha \sqrt(d) \log n$, but should that $d$ on the RHS be $n$?
- Line 171 mentions "Equation (9)" which I can't find.
- Fig 1b goes into margins. In general I don't mind, but out of fairness to other submissions that kept within margins (and had to cut material to do so), I think it was a little unfair. At this point, after acceptance, I don't think it needs to be fixed though (unless someone else complains).